# Structural basis for the synthesis of the core 1 structure by C1GalT1

Andrés Manuel González-Ramírez [1], Ana Sofia Grosso[2,3,7], Zhang Yang [4,7], Ismael Compañón[5,7], Helena Coelho[2,3,7], Yoshiki Narimatsu [4], Henrik Clausen [4], Filipa Marcelo[2,3], Francisco Corzana [5✉] & Ramon Hurtado-Guerrero [1,4,6✉]

C1GalT1 is an essential inverting glycosyltransferase responsible for synthesizing the core 1 structure, a common precursor for mucin-type *O*-glycans found in many glycoproteins. To date, the structure of C1GalT1 and the details of substrate recognition and catalysis remain unknown. Through biophysical and cellular studies, including X-ray crystallography of C1GalT1 complexed to a glycopeptide, we report that C1GalT1 is an obligate GT-A fold dimer that follows a $S_N2$ mechanism. The binding of the glycopeptides to the enzyme is mainly driven by the GalNAc moiety while the peptide sequence provides optimal kinetic and binding parameters. Interestingly, to achieve glycosylation, C1GalT1 recognizes a high-energy conformation of the α-GalNAc-Thr linkage, negligibly populated in solution. By imposing this 3D-arrangement on that fragment, characteristic of α-GalNAc-Ser peptides, C1GalT1 ensures broad glycosylation of both acceptor substrates. These findings illustrate a structural and mechanistic blueprint to explain glycosylation of multiple acceptor substrates, extending the repertoire of mechanisms adopted by glycosyltransferases.

[1] Institute of Biocomputation and Physics of Complex Systems, University of Zaragoza, Mariano Esquillor s/n, Campus Rio Ebro, Edificio I+D, 50018 Zaragoza, Spain. [2] Associate Laboratory i4HB - Institute for Health and Bioeconomy, NOVA School of Science and Technology, 2829-516 Caparica, Portugal. [3] UCIBIO – Applied Molecular Biosciences Unit, Department of Chemistry, NOVA School of Science and Technology, 2829-516 Caparica, Portugal. [4] Copenhagen Center for Glycomics, Department of Cellular and Molecular Medicine, Faculty of Health Sciences, University of Copenhagen, Blegdamsvej 3, DK-2200 Copenhagen N, Denmark. [5] Departamento de Química, Universidad de La Rioja, Centro de Investigación en Síntesis Química, E-26006 Logroño, Spain. [6] Fundación ARAID, 50018 Zaragoza, Spain. [7] These authors contributed equally: Ana Sofia Grosso, Zhang Yang, Ismael Compañón, Helena Coelho. ✉email: francisco.corzana@unirioja.es; rhurtado@bifi.es

In metazoans, mucin-type (GalNAc-type) *O*-glycosylation is initiated by the large and complex family of initiating polypeptide GalNAc-transferases (GalNAc-Ts)[1–3]. These glycosyltransferases (GTs) synthetize the Tn antigen (GalNAc-α-1-*O*-Thr/Ser or α-GalNAc-Thr/Ser)[1–4]. While the addition of α-GalNAc is controlled by twenty different GalNAc-Ts, the elongation of the α-GalNAc (Tn antigen) in all cells is typically determined by a Golgi follow-up inverting galactosyltransferase, termed C1GalT1 or core 1/T-synthase (CAZy31). This GT synthetizes the core 1 disaccharide, also called T antigen (Galβ1-3GalNAc-α-1-*O*-Thr/Ser)[5]. In normal cells, the T antigen is elongated through modification by the addition of other monosaccharides to generate thousands of different *O*-glycans species on glycoproteins[6]. Both the Tn and T antigens are specific human tumor-associated carbohydrate antigens (TACAs) found in clinical specimens of different types of cancer[4,7].

The C1GalT1 is unique among metazoan GTs in that its folding, stability and activity only in higher eukaryotes depends on a private X-linked chaperone Cosmc[8], which interestingly exhibits sequence similarity with C1GalT1 and lacks the catalytic DxD motif[9]. Interestingly, in lower eukaryotes such as *Drosophila melanogaster* or *Caenorhabditis elegans*, C1GalT1 related sequences also exist[5,10], but these enzymes do not appear to require a chaperone for expression[11]. The endoplasmic reticulum Cosmc binds to the unfolded C1GalT1 and is required for its folding[6]. Both C1GalT1 and Cosmc are ubiquitously expressed, which corresponds with the detection of core 1 *O*-glycans structures in most cells[8,12,13]. *C1GalT1* homozygous knockouts (KOs) in mice and *D. melanogaster* exhibit embryonic lethality, with defective angiogenesis and fetal embryonic hemorrhage in mice, and a predominant central nervous system phenotype in *D. melanogaster*, indicating that *O*-glycosylation is essential for normal development and angiogenesis[10,14]. The functions of C1GalT1 and Cosmc have demonstrated that *O*-glycans may conceivably interact with almost all physiological processes, including tissue homeostasis, our immune system homeostasis, the homing and circulation of our blood cells, the protection and integrity of inner and outer epithelial barriers, and maintenance of B cell tolerance[15–17]. Regarding tumorigenesis and metastasis, the Tn antigen is highly expressed in human solid tumors, being one of the most recognized TACAs. In most cases, the Tn antigen is formed due to the hypermethylation of the Cosmc promoter leading to its silencing[18]. Aberrant Tn expression is associated with oncogenic features, including proliferation, migration, and invasion of cancer cells[6,7]. The silencing of Cosmc has also been used to glycoengineer HEK or CHO cells to produce SimpleCell lines allowing the interrogation of the activity of GalNAc-T isoenzymes and analysis of the functions of protein glycosylation[19].

At an enzymatic level, while the GalNAc-Ts exhibit clear preferences for acceptor substrate peptide sequences[3,20–23], it is still unclear to what extent the first elongation step by C1GalT1 involves preferences for the peptide sequence around the GalNAc moieties, and/or the positions and clustering of GalNAc moieties. However, two different studies using the rat and the human C1GalT1 with a series of glycopeptides pointed out that the amino acid sequences around the glycosite finely tune the kinetic parameters of C1GalT1[24,25]. Another study with the *D. melanogaster* C1GalT1 (*Dm*C1GalT1) demonstrated that this GT was active on different glycopeptides although full kinetics experiments were not performed[26]. Nevertheless, the GalNAc *O*-glycoproteome is vast and with enormous sequence variation around glycosites, so it is predicted that C1GalT1 efficiently transfers Gal to all GalNAc moieties (Tn) on proteins indiscriminately and independently of the underlying peptide sequences and clustering of GalNAc *O*-glycans[27]. Overall, the lack of structural information on this enzyme has impeded obtaining mechanistic insights into the glycosyl transfer reaction or an understanding of the molecular basis of the requirement for not only recognition of the GalNAc moiety but also for the surrounding amino acids. Here, we have applied a multidisciplinary approach that has allowed us to uncover the molecular basis of C1GalT1 catalysis and recognition of uridine diphosphate galactose (UDP-Gal) and glycopeptide acceptor substrates. In particular, our data show that C1GalT1 is a GT-A fold dimer that follows the typical $S_N2$ mechanism for inverting GTs with an Asp residue as the catalytic base. The binding of the glycopeptides to the enzyme is mainly driven by the GalNAc moiety while optimal binding and kinetic parameters are reached in the presence of both the GalNAc moiety and the peptide. In addition, we unveil that C1GalT1 recognizes the staggered conformation for the α-GalNAc-Thr linkage, a high-energy conformation that is negligibly populated in solution. With this 3D-arrangement, characteristic of α-GalNAc-Ser peptides, C1GalT1 ensures broad glycosylation of both acceptor substrates.

## Results

**Kinetics of *Dm*C1GalT1 against glycopeptide substrates**. To perform biophysical experiments using *Dm*C1GalT1, we designed a construct that did not contain the predicted signal sequence and the transmembrane domain, and the enzyme was secreted from HEK293 cells (residues T43-Q388; see Supplementary Fig. 1 and *methods*). To assess the activity of *Dm*C1GalT1, we designed a series of glycopeptides (designated P1–P7) based on a previous study[24] (Fig. 1). These glycopeptides contained a Gly at +1 and either a Phe or Tyr at +3, residues that clearly improved the activity, and Tyr/Phe/Pro at −3 that enhanced the activity slightly of the human C1GalT1[24]. Glu at −1 was present in P6 to compare it with the most similar glycopeptides, P1 (Ala at −1) and P7 (Asp at −1). The other positions were occupied by a Pro at +2 and Ala at −2 because both were previously well tolerated[24]. We also included the naked APDTRP, the APDT*RP and the APDS*RP glycopeptides for further evaluation (where * represents a GalNAc moiety bound to the underlying amino acid, either Thr or Ser). The APDTRP is an immunogenic epitope found in the tandem repeat sequence present in MUC1 and the APDT*RP, whose structure in the bound state with an antitumor antibody was recently reported[28], is the basis for development of several cancer vaccines[29], and it is a natural substrate for C1GalT1 in the context of MUC1[28,30]. The use of the APDS*RP was to confirm whether the activity of C1GalT1 was better with a glycopeptide containing α-GalNAc-Thr over α-GalNAc-Ser, as previously reported[25].

To initiate kinetic studies, we set up first the experimental conditions using *Dm*C1GalT1 toward UDP-Gal and APDT*RP (Fig. 1a, Supplementary Fig. 2a and Supplementary Table 1). *Dm*C1GalT1 showed a hyperbolic profile in the presence of variable concentrations of either UDP-Gal or APDT*RP, which was also observed in the presence of the other glycopeptides (Fig. 1a and Supplementary Fig. 2b). The apparent $K_m$s ($K_m^{app}$) for UDP-Gal and APDT*RP were $88 \pm 8$ and $195 \pm 43\,\mu M$, respectively, and the $k_{cat}^{app}$ was ~3.5 min$^{-1}$ (Fig. 1b, left and middle panels, and Supplementary Table 1), a value consistent with other previously reported low $k_{cat}^{app}$ values for follow-up GTs such as POMGnT1/POMGnT2 ($k_{cat}^{app}$ ranged from 7 to 1920 min$^{-1}$ depending on the glycopeptide sequences[31,32]). As expected, *Dm*C1GalT1 was inactive on the naked APDTRP and its activity was slightly reduced in the presence of APDS*RP. Particularly, the $K_m^{app}$, $k_{cat}^{app}$, and catalytic efficiency using APDS*RP were 1.26-, 1.85-, 2.25-fold worse than those of C1GalT1 in the presence of APDT*RP (Fig. 1b), a finding that

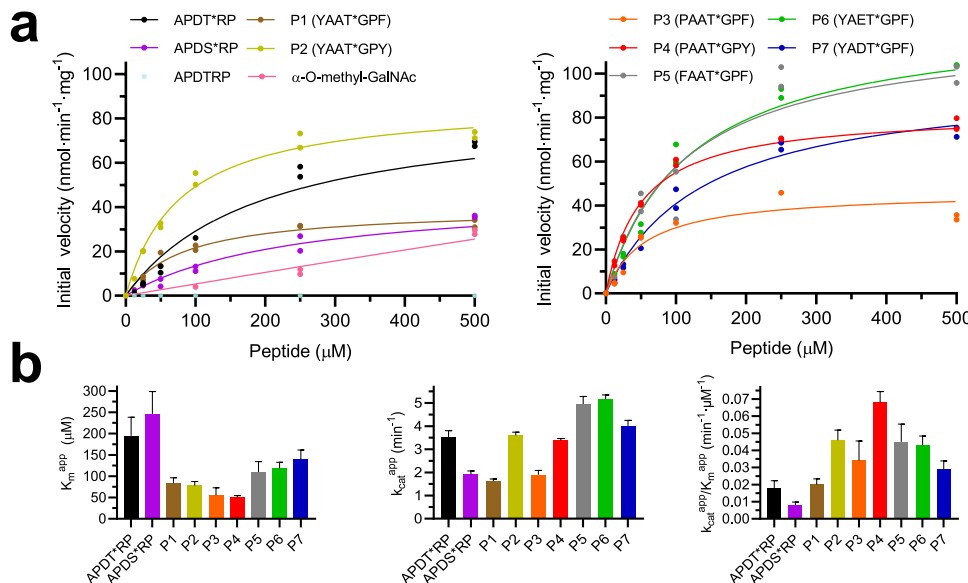

**Fig. 1 Enzyme kinetics experiments of _Dm_C1GalT1[T43-Q388] on (glyco)peptides and α-_O_-methyl-GalNAc. a** Glycosylation kinetics of _Dm_C1GalT1[T43-Q388] against (glyco)peptides and α-_O_-methyl-GalNAc. **b** Plots comparing the $K_m^{app}$, $k_{cat}^{app}$ and the catalytic efficiency ($k_{cat}^{app}/K_m^{app}$) between the different substrates. Additional kinetic data are given in Supplementary Table 1. Initial velocities were obtained in duplicate ($n = 2$ independent experiments) for each peptide concentration. Error bars represent the standard error calculated by the GraphPad Prism fit of the data sets. Source data are provided as a Source Data file.

matches the results found with the rat enzyme[25], and that suggests that the methyl group of Thr is likely important for obtaining slightly better kinetic parameters. We also determined whether the GalNAc moiety behaved as an acceptor substrate. To explore that, we used the α-_O_-methyl-GalNAc as a substrate, which turned out to be a worse substrate than the APDT*RP since kinetic parameters could not be determined, a finding in agreement with a previous report using similar analogues versus glycopeptides[26]. At a saturating concentration of the glycopeptides (~500 μM), the initial activity was ~half of the achieved one with APDT*RP, and the enzyme did not reach saturation up to 2 mM α-_O_-methyl-GalNAc (Fig. 1a and Supplementary Fig. 3). These data show that the context of the peptide around the sugar moiety is key to having optimal kinetic parameters and that the GalNAc moiety is not sufficient to achieve that.

Regarding the kinetics with P1–P7 glycopeptides (Fig. 1a), we firstly determined the kinetic parameters for UDP-Gal under a saturated concentration of P4, rendering a slightly better $K_m^{app}$ and almost an identical $k_{cat}^{app}$ value for UDP-Gal[P4] compared to those parameters for UDP-Gal under the presence of APDT*RP (Supplementary Fig. 2 and Supplementary Table 1). The $K_{ms}^{app}$ for P1–P7 glycopeptides were slightly better than that of the natural APDT*RP glycopeptide, ranging from 1.3- to 4-fold improvements, with P4 having the better $K_m^{app}$ (Fig. 1b, left panel). The data also suggest that the Pro at −3 is slightly better for binding than the Tyr at −3 (P2 versus P4), and that Glu or Asp at −2 are slightly worse for binding than the Ala at −2 (P6/ P7 versus P1). With the $k_{cat}^{app}$ parameters, the range of values is more restricted, with P1 being the slowest substrate and P6 being the fastest (Fig. 1b, middle panel). Finally, the range of catalytic efficiency values were slightly less restricted than that of $k_{cat}s^{app}$, and suggested that for the series of glycopeptides containing an acceptor glycosylated Thr, the best substrate was P4 and the worst ones were P1 and APDT*RP (Fig. 1b, right panel and Supplementary Table 1). Overall, our data suggest that the differences in the kinetic parameters between the glycopeptides are small and that not only the GalNAc moiety is important for glycosylation, but also that the peptide sequence is crucial for

achieving optimal kinetic parameters, suggesting that C1GalT1 may interact directly with the peptide of the acceptor substrates. Note that saturation is not achieved in the presence of α-_O_-methyl-GalNAc and that only this is achieved in the presence of the different peptides within the glycopeptide substrates.

**STD NMR reveals that _Dm_C1GalT1 directly engages with the GalNAc moiety and the peptide sequence.** We then performed STD NMR experiments to shed light onto the _Dm_C1GalT1-glycopeptides interaction mode. The STD NMR experiment is a ligand-observed technique (only the [1]H-NMR assignment of the ligand is required for analysis) that relies on saturation transfer, through nuclear Overhauser effect, from receptor (e.g., protein/ enzyme) proton resonances to protons of a ligand (e.g., carbo-hydrate, glycopeptide) exchanging between a protein-bound and free state[33]. Analysis of the STD responses allows to infer which atoms of the binding ligand are in closer contact with the receptor, and to determine the so-called STD-derived epitope mapping[34]. We selected the α-_O_-methyl-GalNAc, the naked APDTRP, one of the worst substrates (APDT*RP), and one of the best substrates (P4). In the case of α-_O_-methyl-GalNAc, the naked and the APDT*RP, two on-resonance frequencies were used, at aliphatic (−0.5 ppm) and aromatic (7 ppm) region. However, for P4 (due to the presence of the aromatic Tyr), only on-resonance frequency at −0.5 ppm was used. First, in the presence of a ~6-fold and 7-fold excess of UDP and MnCl₂ with respect to the enzyme, we found that while the naked peptide itself did not display STD response, the α-_O_-methyl-GalNAc clearly presented STD enhancements, indicating the importance of GalNAc for the enzyme recognition (Supplementary Figs. 4 and 5a, and Supplementary Table 2). Next, we performed STD NMR of the glycopeptides APDT*RP and P4 in the absence and presence of UDP and MnCl₂. We conducted these experiments in the absence of the nucleotide because previously we found that other distant GT-A fold GTs such as GalNAc-T2, and the NleB/ SseK effector proteins were dependent on the presence of UDP for binding to their protein/peptide acceptor substrates, implying

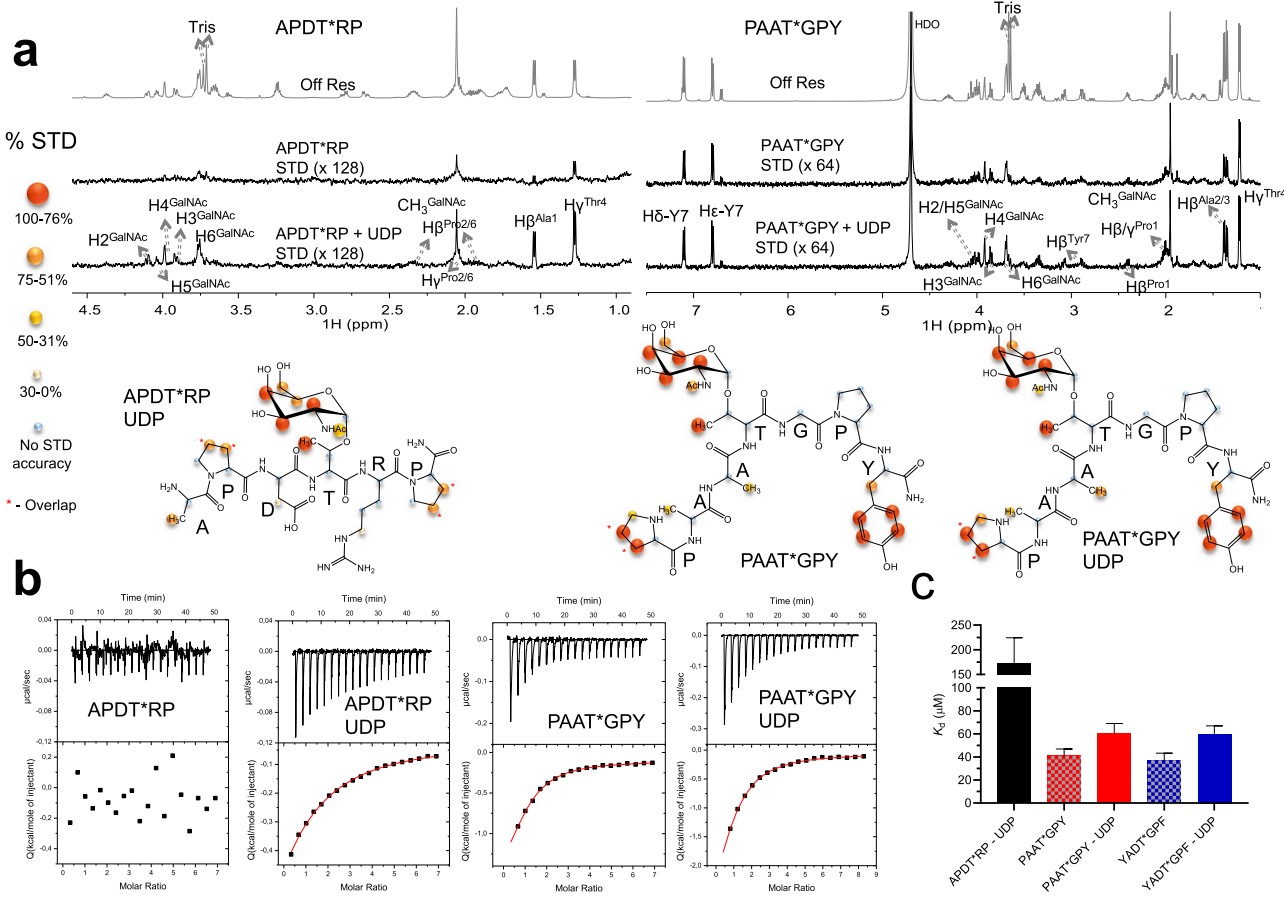

**Fig. 2 STD NMR experiments and ITC experiments. a** Off resonance (Off Res) and STD spectra of APDT*RP and P4 in the absence and presence of UDP (on resonance frequency at −0.5 ppm) (upper panel). STD NMR-derived epitope mapping for APDT*RP and P4 (lower panel). The different colored spheres indicate the normalized STD signal (in %) observed for each proton. The proton resonances that overlap in the spectrum are identified in the figure and displayed with *. **b** ITC data for the binding of APDT*RP and P4 to $Dm$C1GalT1. Top: raw thermogram (thermal power versus time). Bottom: binding isotherm (normalized heats versus molar ratio). The ITCs for APDT*RP and P4 were performed in the absence and presence of an excess of UDP. All our ITCs thermograms have low Wiseman "$c$"-parameters (c < 1), explaining why the initial plateau is missing. Despite this, data reliability is still valid as previously described before[70,71], and in particular the estimated $K_d$s from the low Wiseman "$c$"-parameters are still reliable as the possible error in n estimation has almost no effect in the $K_a$ parameter adjustment. Note that the Wiseman "$c$"-parameter is the product of the receptor concentration and the association constant, $K_a$[63]. The experiments were performed in duplicate ($n = 2$ independent experiments). **c** Graph depicting the $K_d$s for APDT*RP in the presence of UDP, and P4/P7 in the presence or absence of UDP. Error bars represent the error calculated through iteration fit of the data sets by the Origin 7 (Microcal).

the existence of an induced-fit mechanism[20,35,36]. Interestingly, in the case of GalNAc-T2, the active conformation of GalNAc-T2, characterized by the shifting of a flexible loop from an open to a closed conformation, was completely achieved in the presence of UDP-GalNAc and less in the presence of UDP[20]. We also found that the GT-B fold FUT8 showed similar properties to the other two GTs, though FUT8 bound better to an N-glycan in the presence of GDP, and the nucleotide was not essential for binding to the N-glycan[37,38]. In addition, NleB/SseK and FUT8 also contained flexible loops that were ordered in the presence of the nucleotide as we found for GalNAc-T2, implying that the active site adopted an active conformation once that these flexible loops bound to the sugar nucleotide[20,35,36,38]. Overall, we proposed for these enzymes that the binding of the sugar nucleotide was required for binding to the acceptor substrate (optimal binding for FUT8 in the presence of GDP) and in turn for glycosylation. Herein, in the case of $Dm$C1GalT1 and in the absence of UDP, only P4 clearly showed STD signals. However, both glycopeptides showed a clear STD response in the presence of UDP. The results suggest differences in the recognition of the two glycopeptides. In

both cases, the GalNAc moiety displayed high saturation transfer indicating that it should be in closer contact with the enzyme. GalNAc STD-derived epitope for these glycopeptides and for the α-$O$-methyl-GalNAc were comparable, implying a similar binding orientation for GalNAc unit in all structures (Fig. 2a, Supplementary Fig. 5a, b and Supplementary Tables 2–4). However, the STD amplification factor was lower in the case of α-$O$-methyl-GalNAc than that of the APDT*RP or P4 (Supplementary Tables 2–4), which likely reflects the expected lower binding affinity of α-$O$-methyl-GalNAc versus those of the glycopeptides, inferred from the kinetics experiments. Indeed, at the level of the peptide sequence, the Thr methyl group displayed a clear STD response in all cases, while the results varied for the rest of amino acids of both glycopeptides. For the APDT*RP, modest STD enhancements were detected for the methyl of Ala1, and few protons of overlapped Pro2 and Pro6. No STD response was observed for Asp3 protons. Remarkably in the case of P4, significant STD response was found for Pro1 and Tyr7 side chains protons, either in absence or presence of UDP (Fig. 2a), suggesting that these amino acids should be in close contact with

DmC1GalT1. These additional interactions might explain the differences in $K_m$ᵃᵖᵖ between both glycopeptides (fourfold better $K_m$ᵃᵖᵖ of P4 than that of APDT*RP). Overall, the data suggest that the binding of the GalNAc moiety is the driving force for recognition, and optimal binding is reached in the presence of both the GalNAc moiety and the peptide.

**DmC1GalT1 does not show an allosteric behavior with glycopeptides.** To corroborate the different behavior between the glycopeptides in the absence and presence of UDP by STD NMR, we performed isothermal titration calorimetry (ITC) experiments. First, we determined the $K_d$ of UDP for binding to DmC1GalT1 in the presence of $MnCl_2$ ($K_d = 18.39 \pm 4.67 \mu M$) (Supplementary Fig. 6, and Supplementary Table 5). As expected, no binding was shown for the naked APDTRP under an excess of UDP (Supplementary Fig. 6). Then, we evaluated whether this enzyme requires UDP binding prior to binding the glycopeptides. While DmC1GalT1 only showed binding to the APDT*RP in the presence of UDP, DmC1GalT1 bound well to P4 in the presence or absence of UDP, in agreement with the results from the STD NMR experiments (Fig. 2b, c). The $K_d$s for the glycopeptides matched their $K_m$sᵃᵖᵖ and the differences found between the $K_m$sᵃᵖᵖ (~3.5-fold better $K_d$ of P4 than that of APDT*RP). Since the APDT*RP is an unusual glycopeptide containing two charged residues (an Asp and Arg residue), we wondered whether this could be the reason for its behavior in the absence of UDP. To rule out this, we also performed ITC experiments with P7, which contains a negatively charged residue. P7 behaved similarly to P4 and bound indistinctly to the enzyme in the presence or absence of UDP (Fig. 2b, c, Supplementary Fig. 6 and Supplementary Table 5), suggesting that the Arg residue of APDT*RP or its conformation might be behind its behavior (see further experiments below). Regarding the analysis of the thermodynamic parameters of the interaction, these were somewhat complex and difficult to interpret for the glycopeptides, impeding obtaining a meaningful conclusion (Supplementary Table 5).

Our results also imply that DmC1GalT1 does not likely follow an induced-fit mechanism as found for other GTs such as NleB1[36], GalNAc-T2[20,35] and FUT8[38], and that therefore, DmC1GalT1 does not need prior binding to the sugar nucleotide to bind its acceptor substrates.

**Architecture of the DmC1GalT1-UDP-APDT*RP complex.** To provide atomic insights into the structure of DmC1GalT1 and its interaction with UDP-Gal/UDP and glycopeptides, we worked with a truncated version of DmC1GalT1 (residues S73-Q388) that was secreted from High Five (Hi5) cells (see Methods). The kinetic parameters of this construct were highly similar to those found for the longer construct DmC1GalT1ᵀ⁴³⁻Q³⁸⁸ (see Supplementary Fig. 7 and Supplementary Table 1), verifying that the further truncation of N-terminal residues did not affect the kinetic properties. Crystals of the DmC1GalT1 in the presence of UDP-$MnCl_2$ and APDT*RP were obtained in the space group P2₁. Other attempts with other glycopeptides failed to obtain crystals. The crystal structure was obtained at 2.40 Å by molecular replacement and using the DmC1GalT1 model obtained from alpha fold 2 server[39] (Methods, Fig. 3a, upper panel, and Supplementary Table 6). The asymmetric unit (AU) of P2₁ crystals contained two molecules of DmC1GalT1 that were arranged as a homodimer with each monomer adopting the typical GT-A fold (Fig. 3a). The dimeric form was confirmed by gel filtration chromatography (Supplementary Fig. 8b) and was also reported for the human C1GalT1 orthologue[5]. The PISA server further confirmed this dimeric structure and revealed that the dimer presented a large buried surface area (7621 Å²), implying a very

stable and tight interface. The residues at the interface engaged in the stabilization of the dimer were located at the N-terminus loop, α1, α2, α4, loop α6- α7, α7, loop α7-α8, loop α8-β7, β7, loop β7-α9, α9, and the long and unstructured C-terminus loop (Fig. 3a, lower panel, and Supplementary Fig. 1). One of these residues, Tyr321 (336 in DmC1GalT1), highly conserved and located in α9 at the interface, was found mutated to Asn, leading to thrombocytopenia and kidney disease in mice (Supplementary Fig. 1). Interestingly, Cosmc was shown to bind to residues 83–97 of the human C1GalT1 (HsC1GalT1)[40], located in α1, β1 and loop β1-α2 of the DmC1GalT1 structure. One of these residues, Leu95ᴰᵐ C1GalT1 (Leu82ᴴˢ C1GalT1) is in α1 at the dimer interface (Supplementary Fig. 1), suggesting that it is likely that Cosmc is important to form the obligate dimer of C1GalT1. However, this peptide region is partly conserved within C1GalT1 found in vertebrates and invertebrates, implying that this particular peptide in DmC1GalT1 is likely not recognized by Cosmc. Nevertheless, both examples illustrate the importance of interface residues in stability and function of C1GalT1[41]. The root-mean-square deviation (RMSD) between both molecules belonging to chain A and B in the AU is 0.24 Å on 278 equivalent Cα atoms. Hereafter we will discuss only molecule A because it contains a better-defined density for the ligands. In addition, DmC1GalT1 also contained the four conserved landmark features among GT-A GTs[42]: the DxD motif for metal cation interactions (Asp181-X-Asp183), a "glycine-rich" loop facing the acceptor and donor sugar site located in DmC1GalT1 at loop β5-β6, an "xED" motif at the beginning of α6 in DmC1GalT1 harboring the catalytic base (Asp255, see further experiments below), and a "C-His" residue that coordinates with the metal ion (His324) (Fig. 3b and Supplementary Fig. 1).

A close inspection of the active site of DmC1GalT1 and its comparison with other orthologs such as the human, mouse, and chicken C1GalT1 revealed that both the UDP-Gal and glycopeptide binding sites were identical (Fig. 3c, upper panel, and Supplementary Fig. 1), exemplifying that the DmC1GalT1 is an excellent model to understand the biochemical aspects of the human enzyme. An analysis of the electrostatic surface potential showed a negatively charged UDP-Gal binding site required to coordinate the $Mn^{2+}$, and moderate positively charged patches and neutral patches for binding to the peptide. In addition, the GalNAc binding site was moderately negatively and positively charged facing the central core and the acetamide group/OH6, respectively (Fig. 3c, lower panel).

Regarding the structural homology of DmC1GalT1 to other described structures, the DALI server[43] revealed structural homology to two galactosyltransferases, namely the dimeric human B3GNT2 (e.g., PDB entries 7JHN[44] and 6WMO[45]) and the monomeric mouse Manic fringe (Mfng; PDB entries 2J0A and 2J0B[46]), both belonging to the CAZy31 family (Fig. 3d). Although DmC1GalT1 is very distant to B3GNT2 and Mfng in terms of acceptor substrates, the server rendered good scores implying that they superimposed fairly well (RMSDs of ~1.7 and ~3.17 Å between DmC1GalT1 and B3GNT2, and DmC1GalT1 and Mfng crystal structures, respectively; the superimposed residues ranged from 189 to 151 residues). Interestingly, the strong similarities between DmC1GalT1 and B3GNT2 at the overall fold were matched by the excellent superposition of the UDP and the acceptor substrates (Fig. 3e). It is worth mentioning that the GalNAc OH3 of APDT*RP and Gal OH3 of LNnT were located at almost identical positions (~0.92 Å atomic shift between the GalNAc and the Gal moieties) and close to the β-phosphate, in agreement with their role as the acceptor sites. Note that UDP in Mfng also superimposed very well with UDP of DmC1GalT1 though the former structure was only obtained with UDP-$Mn^{2+}$.

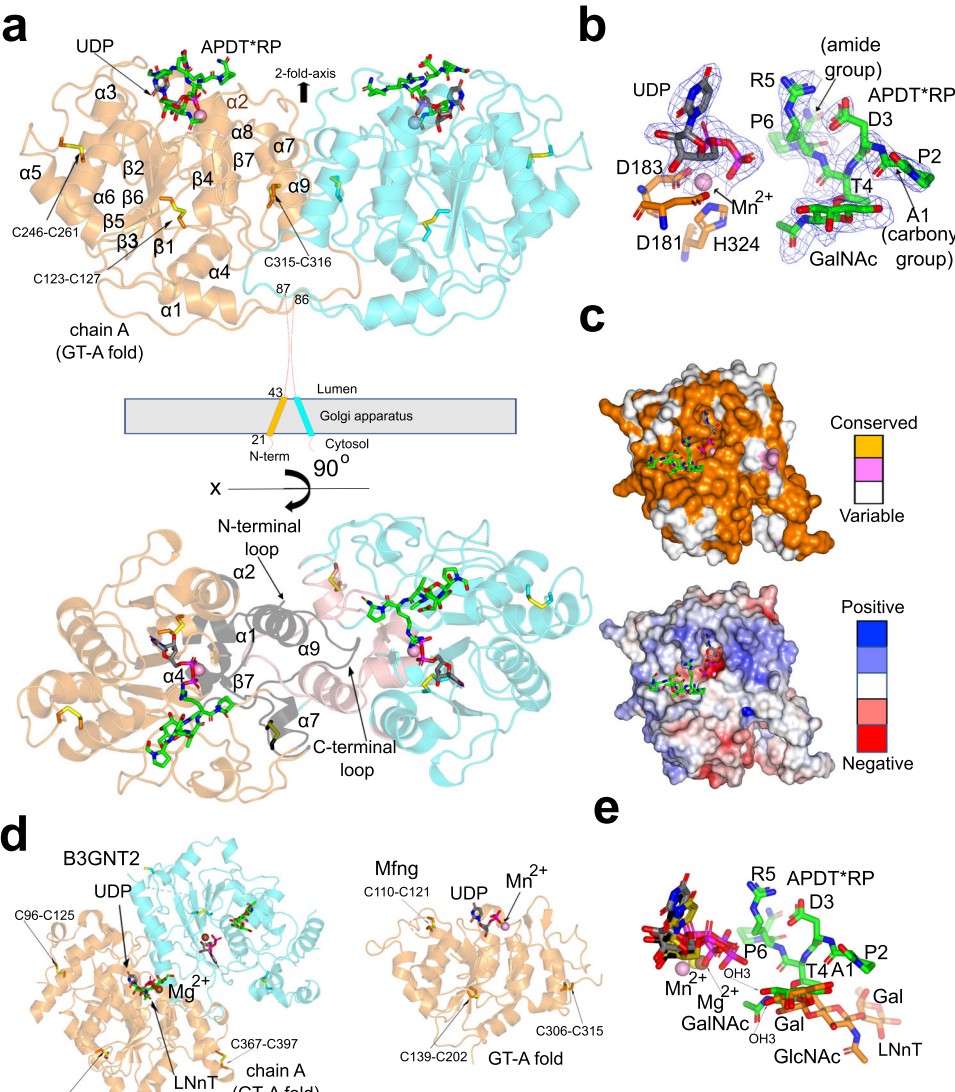

**Fig. 3 Crystal structure of *Dm*C1GalT1$^{S73-Q388}$ in a ternary complex with UDP-Mn$^{2+}$-APDT*RP. a** Ribbon structure of the dimeric form of *Dm*C1GalT1 complexed to UDP-Mn$^{2+}$ and APDTR*P (upper panel). Chain A and B of *Dm*C1GalT1 are colored in orange and cyan, respectively. The UDP nucleotide is depicted with gray carbon atoms whereas the manganese atom is shown as a pink sphere. The glycopeptide is shown as green carbon atoms. Disulfide bridges are indicated as yellow sulfur atoms. Each monomer contains three conserved disulfide bridges, with the latest being formed by two contiguous Cys residues (see also Supplementary Fig. 1). In the lower panel, the dimeric form is displayed in a different view highlighting the secondary structures and the N- and C-terminal loops regions engaged in interactions in the interface (black and pink for chain A and B, respectively). In (**b**) Close-up view of the active site showing the bound Mn$^{2+}$, UDP and ADPT*RP. Electron density maps are Fo–Fc (blue) contoured at 2.2 σ for APDT*RP and UDP. Except for the first N-terminal residue (Ala1) of APDT*RP, the density for the glycopeptide and UDP was well defined. Note that the Pro6 finishes as an amide group. **c** (upper panel) Surface representation of *Dm*C1GalT1, color-coded by degree of sequence conservation. (lower panel) Electrostatic surface representation of *Dm*C1GalT1 (scale bar ranged from −5 kTe$^{-1}$ to +5 kTe$^{-1}$). **d** Structural homologues of *Dm*C1GalT1. Ligands and cofactors follow the same colors as indicated above. **e** Superposition of the ligands of *Dm*C1GalT1, B3GNT2 and Mfng. LNnT and the glycopeptide are depicted as orange and green carbon atoms, respectively. UDP carbon atoms are colored as gray in *Dm*C1GalT1, olive in B3GNT2 and black in Mfng. Mn$^{2+}$ is shown as a pink and yellow sphere in *Dm*C1GalT1 and Mfng, respectively. Mg$^{2+}$ is shown as a yellow sphere.

**The active site of *Dm*C1GalT1.** The *Dm*C1GalT1 binding site is formed by the UDP-Gal and the glycopeptide binding sites (Fig. 4a). The uridine moiety of UDP establishes a CH−π inter-action with Leu155 while the uracil moiety is tethered via hydrogen bonds to Glu150 and Lys158 side chains and Gly151 backbone. The ribose moiety of uridine interacts with the Asp182 side chain and Met160 backbone, and the pyrophosphate interacts with Arg152, His324 and Tyr325 side chains. The pyr-ophosphate group oxygen atoms, Asp181 and Asp183 of the DxD motif and His324 hexagonally coordinate Mn$^{2+}$.

Unlike the intimate recognition of UDP by *Dm*C1GalT1, APDT*RP displays fewer contacts with the enzyme (Fig. 4a), in line with our ITC data in which the binding of UDP was ~9.5-fold stronger than the binding of APDT*RP to the enzyme (Supplementary Table 5). The glycopeptide GalNAc moiety is recognized through hydrogen bonds formed between the acetamide carbonyl and Ser220 side chain, OH3 with Asp255 side chain, OH4 with Asp255/Tyr218 side chains, and OH6 with Tyr304 side chain. At the peptide level, the Pro2 side chain and the methyl group of Thr4 establish CH−π interactions

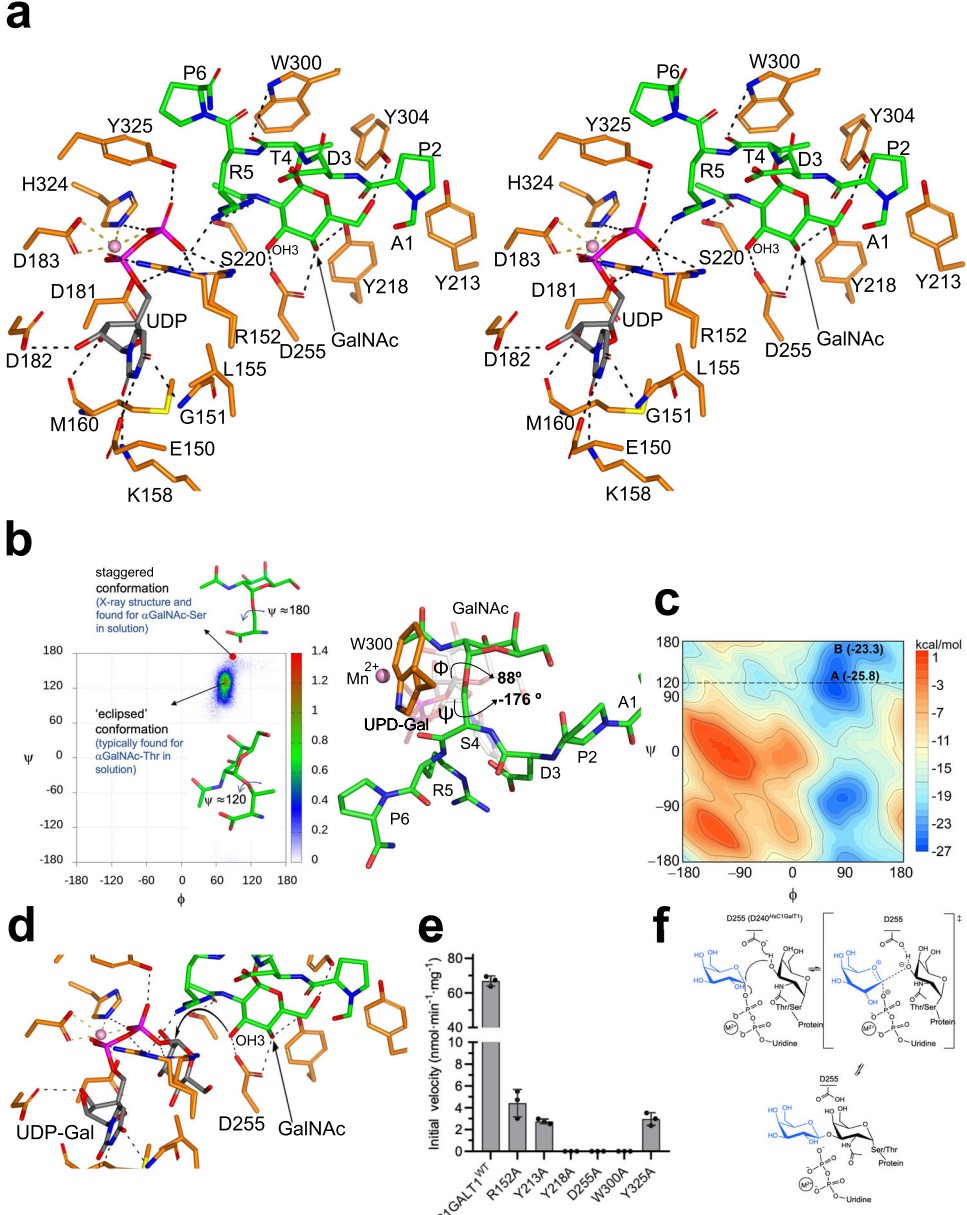

**Fig. 4 Structural features of the active site. a** Stereo view of the active site for the *Dm*C1GalT1-UDP-Mn$^{2+}$-APDT*RP complex. The residues forming the active site are depicted as orange carbon atoms. UDP and the glycopeptide are shown as gray and green carbon atoms, respectively. The manganese atom is shown as a pink sphere. Hydrogen bond interactions are shown as dotted black lines. **b** Geometry of the glycosidic linkage of the glycopeptide APDT*RP in solution derived from 0.5 µs MD simulations. The dihedral angles are defined as follow, φ = O5-C1-O1-Cβ, and ψ = C1-O1-Cβ-Cα. The red circle corresponds to the conformation found for this linkage in the crystal structure of the glycopeptide bound to *Dm*C1GalT1 in the presence of UDP. This conformation is also present in glycopeptide APDS*RP in solution (left panel). 3D view of APDS*RP in complex with *Dm*C1GalT1 in the presence of UDP-Gal obtained from 0.5 µs MD simulations (rigth panel). **c** Free-energy map (φ, ψ) of the glycosidic dihedral angle calculated for the free peptide APDT*RP in water (see *Methods*) at 300 K. The contour maps are drawn with a spacing of 1 kcal/mol. Regions that were never visited by the peptide are shown in dark orange. "A" refers to the 'eclipsed' conformation typically found for α-GalNAc-Thr derivatives in solutions[47,50]. "B" refers to the 'staggered' conformation found for α-GalNAc-Ser derivatives in solution[72]. **d** Close-up view of the binding site region of the *Dm*C1GalT1-UDP-Gal-APDT*RP complex showing the Asp255 as the catalytic base in the plausible $S_N2$ single-displacement reaction mechanism. Note the proximity and the orientation of the GalNAc OH3 to the anomeric carbon (3.81 Å) which is compatible with the inversion of the configuration during the reaction. **e** Histogram showing the relative activities of the mutants compared to the wild-type (WT) protein. All experiments were done in triplicate (*n* = 3 independent experiments). Error bars represent the standard deviation calculated by the GraphPad Prism fit of the data sets. Source data are provided as a Source Data file. **f** Proposed $S_N2$ single-displacement reaction mechanism for C1GalT1.

with Tyr213, and Trp300/Tyr304, respectively, and the Thr4 backbone makes a hydrogen bond with Trp300 side chain. Arg5 side chain is engaged in a hydrogen bond with Arg152, and Pro6 establishes a CH–π interaction with Tyr325 (Fig. 4a). These interactions also reveal that the GalNAc moiety is more intimately recognized than the peptide and that the GalNAc moiety is only tethered through hydrogen bonds, while the peptide is engaged in hydrophobic and hydrogen bond interactions. Overall, our data align with the STD-derived epitope map, suggesting that the GalNAc moiety is the driving force for

recognition and that the peptide improves binding by establishing direct interactions with the enzyme.

**A high-energy conformation of the glycosidic linkage of α-GalNAc-Thr is required for the molecular recognition by *Dm*C1GalT1.** An intriguing feature inferred from the crystal structure was the presence of an energetically less favorable conformation of the glycosidic linkage displayed by α-GalNAc-Thr (Fig. 4b, left panel). This staggered conformation (with $\psi \approx 180°$), typically found in glycosidic linkage between α-GalNAc and a serine residue, is not found in solution for α-GalNAc-Thr either in the free form[47] or bound to proteins, where the eclipsed rotamer (with $\psi \approx 120°$) is the usual form.[21–23,28,35,48–51]. We performed molecular dynamics (MD) simulations on *Dm*C1GalT1 in complex with UDP-Gal and APDS*RP. These calculations showed that the staggered conformation was also predicted for α-GalNAc-Ser (Fig. 4b, rigth panel, and *Methods*), implying that C1GalT1 requires the staggered conformation for the effective glycosylation of α-GalNAc-Ser and α-GalNAc-Thr. MD simulations performed for the analogous complex with APDT*RP, where the eclipsed conformation was fixed in α-GalNAc-Thr (Supplementary Fig. 9 and *Methods*) showed a loss of interactions between the peptide and the protein compared to those found in the X-ray structure. Specifically, the CH-π interactions between the methyl group of Thr4 and Trp300/Tyr304 and Pro6(Cδ) of the glycopeptide and Tyr325 were significantly weakened due to the increased distance between the aromatic rings and the peptide. Moreover, the hydrogen bond between the carbonyl group of Thr4 and Trp300 was negligible throughout the MD simulation trajectory with constraints. As for the GalNAc moiety, the hydrogen bonding between the side chain of Tyr304 and GalNAc OH6 was lost. On the other hand, the APDS*RP peptide has slightly worse $K_m^{app}$ than the threonine derivative and does not have the conformational penalty that operates in the Thr-containing peptide. These results suggest that rather subtle free-energetic effects are probably guiding the binding. In this regard, the free-energy penalty associated to bring the glycosidic linkage from a 'eclipsed' conformation to a 'staggered' one was calculated to be 2.5 kcal/mol (Fig. 4c and Supplementary Fig. 10). In contrast, this conformational shift is favored by 1.9 kcal/mol in the serine derivative. (Supplementary Fig. 10). This finding likely explains why C1GalT1 has similar kinetic parameters for both glycosites, and can glycosylate either α-GalNAc-Ser or α-GalNAc-Thr indistinctly.

**The inversion mechanism of C1GalT1.** To get further insights into the inversion mechanism of C1GalT1, we superimposed our crystal structure with the structure on the human B3GNT2-UDP-GlcNAc complex (PDB entry 7JHL), and then the coordinates of UDP-GlcNAc were replaced by UDP-Gal. The resulting complex, *Dm*C1GalT1-UDP-Gal-Mn²⁺-APDT*RP, was minimized using molecular mechanics (MM) calculation as shown in *Methods* (Fig. 4d). In this structure, the GalNAc OH3 was properly aligned to attack the anomeric carbon atom and compatible with the inversion of the configuration. To confirm the importance of Asp255 for catalysis, we mutated Asp255 to Ala. The activity of the D255A mutant was completely inactive, confirming the Asp255 as the catalytic base (Fig. 4e). Thus, C1GalT1 follows the typical inversion mechanism, in which a catalytic base deprotonates the GalNAc OH3 so the resulting oxyanion can proceed attacking the anomeric carbon of the Gal moiety, which undergoes an oxo-carbenium ion–like transition state (Fig. 4f). Therefore, these results are compatible with an $S_N2$ single-displacement reaction mechanism, which is deployed by most inverting GTs[52].

**In vitro and *in cells* activity of C1GalT1 mutants.** To get insights into the role of residues of *Dm*C1GalT1 engaged in interactions with the glycopeptide, we tested Ala mutations of Arg152, Tyr213, Tyr218, Trp300 and Tyr325 to Ala residues and the resulting mutants were characterized at in vitro level under the same conditions used for the D255A. The results showed that Y218A and W300A were inactive while R152A and Y213A/Y325A suffered a 15- and 25-fold decrease in activity with respect to the WT, respectively (Fig. 4e). We then generated the equivalent mutants of *Dm*C1GalT1 in the *Hs*C1GalT1 (see Supplementary Fig. 1). To evaluate the activity of these *Hs*C1GalT1 mutants in cells, we used a HEK293^Tn cell without capacity for producing core 1 (KO *C1GALT1*) and without capacity to modify the core 1 (T) *O*-glycan, including capacity for core 2 (KO *GCNT1*) and sialylation of core 1 (KO *ST3GAL1/2* and *ST6GALNAC2/3/4*). This cell line would thus have no competitive enzymes working on the Tn *O*-glycan substrate or enzymes converting the T *O*-glycans when produced (Fig. 5a). We then installed the full coding construct of *Hs*C1GalT1 and mutants (R140A, Y201A, Y206A, D240A, W285A and Y310A) by targeted knock-in (KI) (Supplementary Fig. 11 and Fig. 5d). The induction of core 1 (T) expression on cell surface was evaluated by flow cytometry with the anti-T monoclonal antibody (mAb) 3C9 (Fig. 5b, c). mAb 3C9 did not bind HEK293^Tn cell but strongly bound the cells after KI of WT *Hs*C1GalT1. KI of *Hs*C1GalT1 mutants R140A, Y201A, Y206A and Y310A produced partial restoration of 3C9-binding with Y206A being the least effective, while KI of D240A and W285A mutants produced no binding suggesting these were completely inactive (Fig. 5c). Therefore, our results support that the D240 in *Hs*C1GalT1 (D255 in *Dm*C1GalT1) is the catalytic base, and the Y206 (Y218A^*Dm*C1GalT1) and W285 (W300^*Dm*C1GalT1) residues are also critical in recognition and catalysis. Overall, the results in cells with the *Hs*C1GalT1 mutants match those found with the *Dm*C1GalT1 mutants, validating that the *Dm*C1GalT1 enzyme serves as a model for the human enzyme.

**Putative 3D structures derived from Molecular dynamics (MD) simulations.** We generated putative 3D structures for the apo form of the enzyme, as well as for the enzyme in the presence of UDP-Gal and for complexes between *Dm*C1GalT1 and the glycopeptides APDT*RP, APDS*RP, P2, P4, and P7 (Fig. 6, Supplementary Figs. 12–16 and *Methods*). According to these calculations, the protein retains its 3D structure almost unchanged in the presence of UDP-Gal and upon the formation of the ternary complex with APDT*RP (Fig. 6a), consistent with the lack of an induced-fit mechanism. In all complexes, the hydrogen bonds between the GalNAc moiety and the enzyme present in the X-ray structure were observed in the MD simulations, regardless of the peptide sequence (Supplementary Figs. 12–15). Moreover, the glycosidic linkage of all glycopeptides exhibited a staggered conformation which could be a mechanism used by the enzyme to glycosylate α-GalNAc-Thr and α-GalNAc-Ser residues in a similar manner (Supplementary Fig. 14). For the peptide APDS*RP (Supplementary Fig. 16), the calculations show the absence of a CH-π interaction between Trp300 and Ser4. However, a similar interaction was observed between the hydrogen atoms of Cβ of this residue and the side chain of Tyr304. For APDT*RP in complex with *Dm*C1GalT1, the GalNAc and UDP-Gal showed the correct orientation, with a distance O3-GalNAc/C1-Gal <5.5 Å throughout the entire trajectory, which is consistent with the inversion mechanism (Fig. 6b, c and Supplementary Fig. 15). In addition, the binding mode for the glycopeptide observed by MD

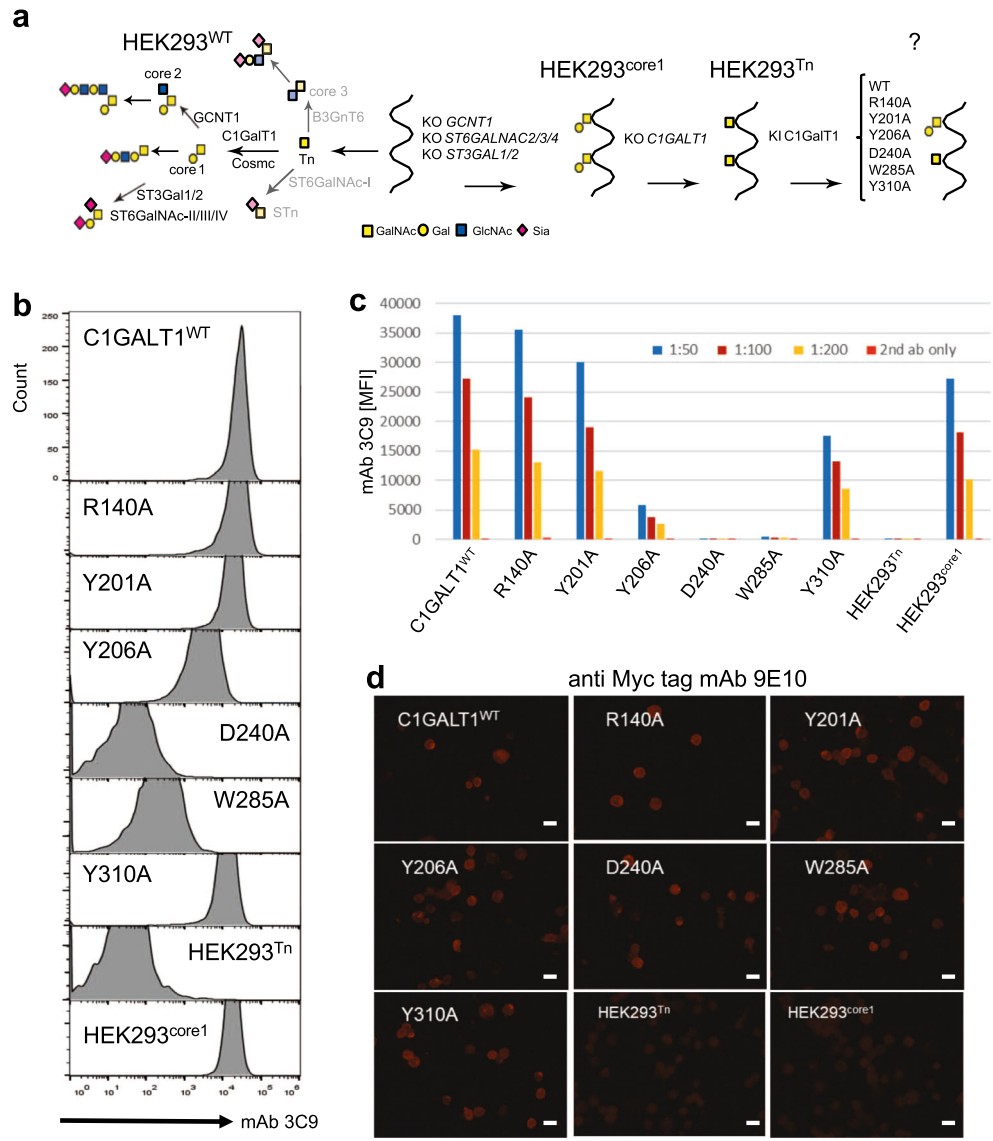

**Fig. 5 Flow Cytometry Analysis of the reinstallation of T glycoform with *Hs*C1GalT1 mutants. a** The *O*-glycosylation pathway is indicated with the name of the enzymes involved in the synthesis of *O*-glycan structures. Note that the non-expressed genes and the predicted basic glycan features missing in HEK293 cells are faded out based on RNA-seq analysis. The engineering strategy to develop HEK293^Tn clone was performed with *C1GALT1* gene KO in HEK293^core1 cells (HEK293^KO GCNT1,ST3GAL1/2,ST6GALNAC2/3/4) followed by the individual KI of C-terminal Myc tagged *Hs*C1GalT1 WT and 6 mutants complementary DNA. **b** Flow cytometry analysis with the core 1 specific monoclonal antibody 3C9 (1 to 100 diluted hybridoma supernatant) were used to evaluate the cell surface level of T or core 1 glycoform. **c** Bar diagrams show mean fluorescence intensities of the mAb 3C9 binding. **d** Immunocytology analysis of single KI clones with anti-Myc-tag mAb 9E10 detecting C-terminal Myc tag of C1GalT1. Note that HEK293^core1 clone (HEK293^KO GCNT1,ST3GAL1/2,ST6GALNAC2/3/4) has endogenous expression of *Hs*C1GalT1 without Myc tag. Images are representative of three experiments (*n* = 3 independent experiments). Scale bar = 20 μm.

simulations agrees with the STD experiments described above (Supplementary Table 7). The absence of UDP-Gal does not significantly alter the interactions between the glycopeptides and the enzyme compared to the ternary complexes, except for the glycopeptide APDT*RP, which agrees with the experimental results. In absence of UDP-Gal, Arg5 of the peptide interacts with Glu254, which leads to a shift of the GalNAc unit from its binding site. Indeed, some frames of the MD simulations of the binary APDTRP-*Dm*C1GalT1 complex show a lack of hydrogen bonds between OH3 and OH4 of the sugar and Asp255 (Fig. 6d). Therefore, the occurrence of UDP-Gal in this complex may stabilize the positive charge and hinder the interaction of Arg5 with Glu254. On the contrary, the absence of UDP-Gal may favor nonspecific interactions with the protein, explaining the absence of binding of this glycopeptide to the enzyme when UDP is not

added. For glycopeptide P2, the MD simulations show three relevant interactions between the peptide fragment and the protein (Supplementary Fig. 13). A hydrogen bond between the side chain of Trp300 of the protein and the carbonyl group Gly5 is present for about 94% of the trajectory. Moreover, the side chains of Tyr231 and Phe299 are involved in CH-π interactions with the N- and C-terminal residues of the peptide, respectively. Similarly, P7 forms a hydrogen bond between its Gly and Trp300, as well as a CH-π interaction between its N-terminal residue and Tyr213 (Supplementary Fig. 13). Finally, the simulations of P4 in complex with UDP-Gal and *Dm*C1GalT1 indicate a highly populated hydrogen bond between Gly5 and the side chain of Trp300 (population ≈ 95%), together with stabilizing contacts between the protein and both the N- and C-terminal regions of the peptide. Also, in the case of glycopeptide P4, good agreement is observed

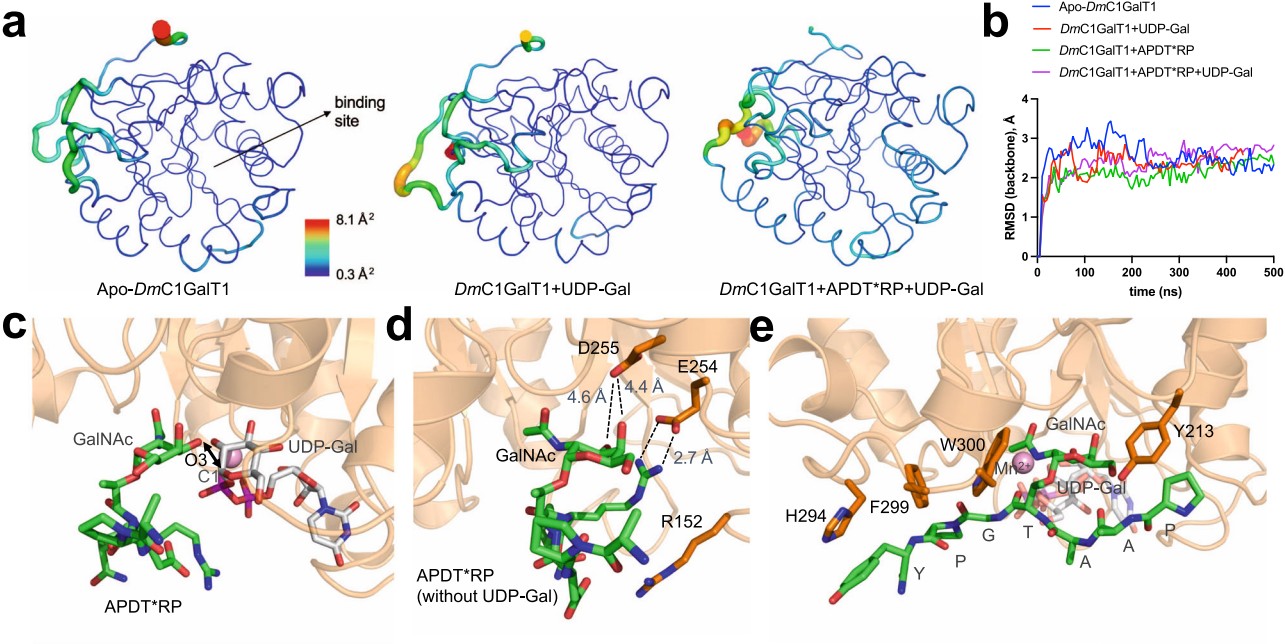

**Fig. 6 MD simulations analysis. a** Atomic fluctuation (Cα) derived from 0.5μs MD simulations for *Dm*C1GalT1 and two binary complexes. The data correspond to the average structure of the protein throughout the simulations. **b** RMSD plots derived from 0.5 μs MD simulations for the apo form and different complexes. Only the backbone atoms of the protein were used for these calculations. **c** Close-up view of *Dm*C1GalT1-UDP-Gal-APDT*RP complex. **d** Close-up view of the *Dm*C1GalT1-APDT*RP complex in the absence of UDP-Gal. **e** Close-up view of the *Dm*C1GalT1-UDP-Gal-P4 complex, showing the interactions between the protein and the N- and C-terminal regions of the peptide fragment.

between the glycopeptide-protein interproton distances derived along the MD simulations in the presence of UDP-Gal and the STD responses estimated for GalNAc, Pro1, Ala2, and Thr4 (Supplementary Table 8). Transient close contacts between Ala3 or Tyr7 with protein residues were observed throughout the MD trajectory, which could also explain the STD response for these amino acids.

## Discussion

The C1GalT1 is critical for the immediate elongation and processing of GalNAc-type protein *O*-glycosylation in most normal cells, and here we provided insights into this enzyme and its catalytic mechanism by solving the crystal structure of the *Drosophila* orthologue. The presence of a private chaperone has been attributed to the fact that the higher eukaryotes C1GalT1 are not N-glycosylated (the lower eukaryotes C1GalT1 are N-glycosylated[11]; see Supplementary Fig. 8a). Yet, this may not necessarily be the explanation because several human GTs lacking N-glycosites are still properly folded without the need for a chaperone[21,35]. We hypothesize here, based on our structural analysis, that Cosmc is likely important in C1GalT1 dimer interface formation in higher eukaryotes. Our results also provide an explanation for the conundrum that the first step in *O*-glycosylation is covered by the largest isoenzyme family catalyzing a single glycosidic linkage presumably to cover the wide variation in substrate sequences in the proteome, while the immediate next step in elongation is covered by only a single non-redundant enzyme, the C1GalT1. C1GalT1 was found to have very broad acceptor substrate specificity and clearly showed the strongest interactions with the GalNAc acceptor sugar residue. Interactions of C1GalT1 with the peptide were identified with some sequence preferences, but these were shown not to be critical for activity. This suggests that the C1GalT1 can serve widely in core 1 *O*-glycan elongation and

cover the entire spectrum of *O*-glycans distributed in the proteome. Clearly, C1GalT1 may have different kinetic properties for GalNAc glycosylated *O*-glycopeptides, but in normal cells, most if not all *O*-glycans are elongated to mask exposure of the cancer-associated Tn structure. Exposure of Tn in cancer cells is generally not due to inactivating mutations in the *C1GalT1* gene[18] and heterogeneous with both Tn and core 1 structure are found in most cancer cells[53]. Thus, reduced expression of C1GalT1 may instead lead to incomplete *O*-glycan elongation with preferences for *O*-glycan sites that are less preferred substrates for C1GalT1.

Core 1 *O*-glycan structures synthesis in human cells also depend on the expression and equilibrium between C1GalT1 and other GTs such as core 3 synthase (B3GnT6) and ST6GalNAc-I (Fig. 5a). While core 3 synthase adds GlcNAc onto the initial GalNAc OH3, ST6GalNAc-I transfers sialic acid onto the Tn antigen GalNAc OH6 forming the STn antigen. In most human cells, the core 1 *O*-glycan structure is the most abundant precursor for building complex *O*-glycans. However, e.g., in normal colon, the major *O*-glycan core structure is the core 3 structure, while interestingly goblet cells also accumulate acetylated STn intracellularly[54,55]. The core 3 synthase is up-regulated in colonic cells while the C1GalT1 is also expressed, and competition for the initial GalNAc residues attached may be in favor of core 3. An explanation for the accumulation of acetylated STn glycoforms intracellularly in goblet cells is less obvious. However, ST6GalNAc-I is selectively expressed in the colon and can compete with C1GalT1, and C1GalT1 cannot transfer to STn *O*-glycans. Our structural studies provide a molecular basis for why C1GalT1 cannot glycosylate the STn antigen. The sialic acid will likely clash with Tyr201 (Tyr213[*Dm*C1GalT1]), Tyr206 (Tyr218[*Dm*C1GalT1]) and Tyr289 (Tyr304[*Dm*C1GalT1]) of *Hs*C1GalT1 (see Fig. 4a and particularly the position of the GalNAc OH6).

Different crystal structures of initiating GTs with acceptor substrates have revealed that these enzymes employ different strategies to recognize their protein substrates[23,36,56]. However, for follow-up GTs acting immediately after the first monosaccharide is attached to the protein backbone, only the crystal structure of POMGnT1 has been published, revealing that this enzyme tethers the mannose moiety through hydrogen bond interactions while the peptide is exclusively recognized by hydrophobic interactions with the enzyme[57]. Herein, the integration of X-ray crystallographic data, STD NMR and molecular modeling allowed to decode the recognition of APDT*RP by *Dm*C1GalT1. From the visual inspection of the complex's crystal structure, the APDT*RP is mainly recognized through GalNAc unit by a network of H-bonds involving OH3, OH4 and OH6. This observation is complemented by the STD NMR spectra of APDT*RP in presence of *Dm*C1GalT1, which provides information about GalNAc aliphatic protons, and pinpoints that GalNAc protons are those in closer contact with the protein, reinforcing the conclusion that GalNAc is the main contact point to the enzyme. With respect to the peptide, both techniques indicate that Pro residues and the methyl group of Thr are involved in the recognition, helping to stabilize the peptide by a mix of hydrophobic and hydrogen bond interactions.

Our structural studies also show the striking finding that the enzyme imposes a non-natural staggered conformation to α-GalNAc-Thr linkage that is typically found in α-GalNAc-Ser. In doing so, α-GalNAc-Thr behaves highly similar to α-GalNAc-Ser except for the Thr methyl group, whose gain in binding through interaction to neighboring active site aromatic residues might compensate for the energy penalty due to the unfavorable conformation for the α-GalNAc-Thr. This feature, which is essential to achieve glycosylation, is likely behind why this enzyme indistinctly glycosylates both acceptor glycosites. It is tempting to speculate that it might be likely more structurally and energetically advantageous for C1GalT1 to impose the staggered conformation to α-GalNAc-Thr, which is low populated in solution, than to adapt its active site to the main conformer found in solution for α-GalNAc-Ser (staggered conformation) and α-GalNAc-Thr (eclipsed conformation). A similar unfavorable enzyme-induced acceptor substrate conformation has been reported for FUT8, in which the enzyme also imposes a more unstable anti-ψ conformation to the core-chitobiose GlcNAc moieties of the N-glycan to achieve core-fucosylation[38]. This clearly exemplifies that enzymes do not always select for more stable acceptor substrate conformations to achieve catalysis and that in cases like the C1GalT1 or FUT8, a more unstable conformation is selected for catalysis.

In summary, we propose that C1GalT1 follows the typical $S_N2$ mechanism described for inverting GTs, and reveal the molecular basis of glycopeptide recognition. We also uncover that C1GalT1 imposes a high-energy and unfavorable conformation to α-GalNAc-Thr as a required step for glycosylation. This is a remarkable example of how GTs have implemented strategies to promote conformational changes in the acceptor substrates to achieve glycosylation.

## Methods

**Production of *Dm*C1GalT1-expressing baculovirus.** The DNA sequence encoding amino acid residues of the *Dm*C1GalT1 (aa S73-Q388) with the mellitin honey bee secretion signal was codon optimized and synthesized by GenScript (USA) for expression in insect cells. The DNA, containing at the 5′-end a recognition sequence for BamHI, and at the 3′-end a sequence encoding for a 6xHis tag, a stop codon and a recognition sequence for EcoRI, was cloned into a pFastBac1, rendering the vector pFastBac1-mellitin-*Dm*C1GalT1-6His. The cloning of the construct into the pFastBac1 was also performed by GenScript.

Recombinant bacmid was produced with the Tn7 transposition method in DH10Bac according to the Bac-to-Bac® Expression System (Invitrogen™).

pFastBac1-mellitin-*Dm*C1GalT1-6His was transformed into *E. coli* DH10Bac cells containing the baculovirus genome (Bacmid DNA). Transposition between the vector and the bacmid occurred through the Tn7 transposition method to generate a recombinant bacmid with the mellitin-*Dm*C1GalT1-6His construct. DH10Bac cells were grown for at least 48 h at 37 °C in LB agar plates containing 50 µg/mL kanamycin, 7 µg/mL gentamicin, 10 µg/mL tetracycline, 100 µg/mL Bluo-gal, and 40 µg/mL IPTG (isopropyl 1-thio-ß-D-galactopyranoside). Positive clones (white colonies) were selected thanks to the disruption of the lacZ gene integrated between the transposition site. The recombinant bacmid was isolated using the NucleoBond BAC 100 (MACHEREY-NAGEL) extraction kit according to the manufacturer's instructions.

All insect cells, *Spodoptera frugiperda* (Sf9) and *Trichoplusia ni* (High Five™ or for simplicity Hi5) (both strains were purchased from GIBCO), were grown in suspension at 27 °C in an incubator with rotation at 130 rpm. P0 baculovirus was produced by transfection of the recombinant bacmid into low-passage suspension Sf9 cells. For the transfection of suspension cells, Sf9 cells were diluted in Insect-XPRESS™ protein free media (LONZA) to 0.8 × 10^6 cells/ml 3–4 h prior transfection. 1 µg bacmid DNA/ml culture was diluted in 100 µl of prewarmed PBS and mixed vigorously with PEI-MAX at a 1:4 ratio of DNA:PEI. DNA and PEI complex formation was allowed for 20–30 min at room temperature and the solution was added afterwards to the cells[58]. The transfected culture was incubated at 27 °C at 130 rpm for 7 days and then harvested by spinning down the cells at 4000 × g for 10 min. The resulting supernatant containing the P0 virus was storage in dark after adding 10% of inactivated FBS.

For both P1 and P2 virus amplification Sf9 cells were diluted to 1.5 × 10^6 cells/ml before adding 0.25% of the prior virus. For the P1 virus amplification the cells were incubated for 7 days while for the P2 amplification the incubation time was 5 days. The incubation, harvesting and storage of the P1 and P2 virus was carried out in the same way as described above with the P0.

**Expression and purification of *Dm*C1GalT1 in insect cells.** For protein expression, High Five™ cells were diluted to 1.5 × 10^6 cells/ml in fresh Insect XPRESS media. At the time of infection, Kifunensine-Bio-X (CarboSynth) was added to the culture (5 µM final concentration to facilitate trimming of N-glycans during the purification, see below), followed by 3% of P2 baculovirus. Cells were harvested 2 days post-infection by spinning down at 300 × g for 5 min, after which the supernatants were collected and centrifuged at 8000 × g for 15 min. Supernatant was dialyzed against buffer A (25 mM TRIS pH 7.5, 300 mM NaCl) and loaded into a His-Trap Column (GE Healthcare). Protein was eluted with an imidazole gradient in buffer A from 10 mM up to 400 mM. Buffer exchange to 25 mM MES pH 6.2, 150 mM NaCl (buffer B) was carried out using a HiPrep 26/10 Desalting Column (GE Healthcare). Endoglycosidase-H (Endo-H) was then added in a ratio 3:250 (Endo-H:protein) in order to trim the N-glycans. After 20 h of reaction at 18 °C, the cleavage was properly verified through SDS-PAGE. ENDO-H was later removed from the solution using a MBP-Trap Column (GE Healthcare), and isolated *Dm*C1GalT1-His was then loaded in HiLoad 26/60 Superdex 75 Colum (GE Healthcare), previously equilibrated with 25 mM TRIS pH 7.5, 150 mM NaCl (buffer C). Quantification of protein was carried out by absorbance at 280 nm using its theoretical extinction coefficient ($\varepsilon_{280nm}$ = 62800 M$^{-1}$ cm$^{-1}$).

**Expression and purification of *Dm*C1GalT1 in mammalian cells.** The DNA sequence encoding amino acid residues of the *Dm*C1GalT1 (aa T43-Q388) was codon optimized and synthesized by GenScript (USA) for expression in HEK293 cells (GIBCO). The construct, containing at the 5′-end a recognition sequence for AgeI, and at the 3′-end a recognition sequence for KpnI, was cloned into a pHLSec plasmid that contained a sequence encoding a 6xHis tag at the 3′-end followed by a stop codon, rendering the vector pHLSec-*Dm*C1GalT1-6His. The cloning of the construct into pHLSec was performed by GenScript. All mutants in *Dm*C1GalT1 (R152A, Y213A, Y218A, D255A, W300A and Y325A) were generated following a standard site-directed mutagenesis protocol by GenScript using the vector pHLSec-*Dm*C1GALT1-6His. pHLSec-*Dm*C1GalT1-6His and all the plasmids encoding for the different mutants were transfected into HEK293 cell line as described below. For DNA amplification, all plasmids were transformed in *E. coli* DH5α cells and extracted with PureLink™ Expi Endotoxin-Free Giga Plasmid Purification Kit (Invitrogen) according to the manufacturer's instructions. Cells were grown in suspension in a humidified 37 °C and 8% CO2 incubator with rotation at 125 rpm. Transfection was performed at a cell density of 2.5 × 10^6 cell/mL in fresh media F17 serum-free media (Gibco) with 2% Glutamax and 0.1% Kolliphor® P 188. For each 150 mL of culture, 450 µg of the plasmid (1 µg/µL) was mixed with 135 µL of sterilized 1.5 M NaCl. This mixture was added to each 150 mL cell culture flask and incubated for 5 min in the incubator. After that, 1.35 mg of PEI-MAX (1 mg/mL) was mixed with 135 µL sterilized 1.5 M NaCl and subsequently the mix was added to the cell culture flask. Cells were diluted 1:1 with pre-warmed media supplemented with valproic acid 24 h post-transfection to a final concentration of 2.2 mM. Then, cells were harvested 6 days post-transfection by spinning down at 300 × g for 5 min, after which the supernatants were collected and centrifuged at 4000 × g for 15 min. Supernatant was dialyzed against buffer A and loaded into a His-Trap Column (GE Healthcare). Protein was eluted with an imidazol gradient in buffer A from 10 mM up to 400 mM. Buffer exchange to buffer C was carried out using a HiPrep 26/10 Desalting Column (GE Healthcare). The isolated

*Dm*C1GalT1-His and its mutants were then loaded in HiLoad 26/60 Superdex 75 Colum (GE Healthcare), previously equilibrated with buffer C. Quantification of the proteins was carried out by absorbance at 280 nm using their theoretical extinction coefficient ($\varepsilon_{280\ nm}$ values ranging between 58790 $M^{-1}\ cm^{-1}$ and 64290 $M^{-1}\ cm^{-1}$ depending on the protein).

**Crystallization and data collection.** Crystals of the *Dm*C1GalT1$^{S73-Q388}$ were grown by sitting drop experiments at 18 °C by mixing 0.5 µL of protein solution (15 mg/mL *Dm*C1GalT1$^{S73-Q388}$, 5 mM UDP, 2 mM MnCl$_2$ and 5 mM APDT*RP in buffer C) with an equal volume of a reservoir solution (0.1 M potassium thiocyanate, 30% Polyethylene glycol monomethyl ether 2000). The crystals were cryoprotected in mother liquor containing 10% glycerol and flash frozen in liquid nitrogen.

**Structure determination and refinement.** Diffraction data was collected on synchrotron beamline I24 of the Diamond Light Source (Harwell Science and Innovation Campus, Oxfordshire, UK) at a wavelength of 0.97 Å and a temperature of 100 K. Data were processed and scaled using XDS[59] and CCP4[60,61] software packages. Relevant statistics are given in Supplementary Table 6. The crystal structure was solved by molecular replacement with Phaser[60,61] using the *Dm*C1GalT1 model obtained from alpha fold 2 server[39]. Initial phases were further improved by cycles of manual model building in Coot[62] and refinement with REFMAC5[63]. The final structure of the *Dm*C1GalT1$^{S73-Q388}$-UDP-Mn$^{2+}$-APDT*RP complex was validated with PROCHECK, model statistics are given in Supplementary Table 6. The AU of the P2$_1$ crystal contained two molecules of *Dm*C1GalT1. The Ramachandran plot for the *Dm*C1GalT1$^{S73-Q388}$-UDP-Mn$^{2+}$-APDT*RP complex shows that 87.8%, 11.2%, 1.0% and 0.0% of the amino acids are in most favored, allowed, generously allowed and disallowed regions, respectively.

**Isothermal titration microcalorimetry (ITC).** ITC was used to characterize the interaction of *Dm*C1GalT1$^{T43-Q388}$ with UDP, APDT*RP with and without UDP, APDTRP with UDP, and P4/P7 with and without UDP. All experiments were carried out in an Auto-iTC200 (Microcal, GE Healthcare) at 20 °C. The titration of *Dm*C1GalT1$^{T43-Q388}$ with UDP was carried out at 100 µM of the enzyme with 800 µM UDP in 25 mM TRIS pH 7.5, 150 mM NaCl and 1 mM MnCl$_2$. The titrations with APDT*RP and P4/P7 in the absence of UDP were carried out at 60 µM of the enzyme with 2 mM P4/P7 in 25 mM TRIS pH 7.5, 150 mM NaCl and 1 mM MnCl$_2$. To determine the $K_{dS}$ for APDT*RP, APDTRP and P4/P7 under an excess of UDP, the experiments were made in 25 mM TRIS pH 7.5, 150 mM NaCl, 1 mM UDP and 1 mM MnCl$_2$. The concentration of the enzyme was 60 µM for the titrations of APDT*RP and APDTRP and 50 µM for the titration with P4/P7. The concentration of the (glyco)peptides was 2 mM in all the ITC experiments. The experiments were performed in duplicate. Data integration, correction and analysis were carried out in Origin 7 (Microcal). The data were fit to a one-site equilibrium-binding model. Stoichiometry (n) of binding in all cases was ~1:1 except for UDP whose $n = 0.4$.

**Kinetic analysis.** Enzyme kinetics for the *Dm*C1GalT1$^{T43-Q388}$, *Dm*C1GalT1$^{S73-Q388}$ and the mutants were determined using the UDP-Glo luminescence assays (Promega). Reactions contained 500 nM of the enzymes in 25 mM TRIS pH 7.5, 150 mM NaCl, 50 µM MnCl$_2$, 1 mg/ml BSA (bovine serum albumin) and 500 µM UDP-Gal in the presence of variable concentrations of the peptides and α-O-methyl-GalNAc. The concentrations of P1–P7 and the α-O-methyl-GalNAc ranged from 12.5 to 500 µM and from 125 µM to 2 mM, respectively. The concentrations of APDT*RP and APDS*RP ranged from 12.5 to 1000 µM to get a better estimation of the non-linear Michaelis–Menten fitting. In order to determine the kinetic parameters for UDP-Gal using *Dm*C1GalT1$^{T43-Q388}$, we used 500 nM *Dm*C1GalT1$^{T43-Q388}$ and variable concentrations of UDP-Gal (12.5 µM–1 mM) in the presence of P4 and APDT*RP at a saturated concentration (250 µM and 1 mM respectively, which was approximately fivefold higher than the $K_m^{app}$ value). For the mutants, the activity assay was performed using the mutants at 500 nM with 500 µM UDP-Gal and 500 µM APDT*RP. Reactions were incubated 30 min at 37 °C and stopped using 5 µl of UDP-detection reagent at a 1:1 ratio in a white and opaque 384-well plate. Then, the plates were incubated in the dark for 1 h at room temperature. Subsequently, the values were obtained by using a Synergy HT (Biotek). To estimate the amount of UDP produced in the glycosyltransferase reaction, we created a UDP standard curve. The values were corrected against the UDP-Gal hydrolysis and were fit to a non-linear Michaelis–Menten program in GraphPad Prism 8 software from which the $K_m^{app}$, $k_{cat}^{app}$ and $k_{cat}^{app}/K_m^{app}$ along with their standard errors were obtained. All experiments were performed in duplicate except for the determination of the activity of the mutants that were performed in triplicate.

**Solid-phase peptide synthesis (SPPS).** (Glyco)peptides were synthesized by stepwise microwave assisted solid-phase synthesis on a Liberty Blue synthesizer using the Fmoc strategy on Rink Amide MBHA resin (0.1 mmol). Fmoc-Thr[GalNAc(Ac)$_3$-α-D]-OH or Fmoc-Ser[GalNAc(Ac)$_3$-α-D]-OH (2.0 equiv) were synthesized as before[64] and manually coupled using HBTU [(2(1H-benzotriazol-1-yl)-1,1,3,3-tetramethyluronium hexafluorophosphate], while all other Fmoc amino acids (5.0 equiv.) were automatically coupled using oxyma pure/DIC (N,N′-diisopropylcarbodiimide).

The O-acetyl groups of GalNAc moiety were removed in a mixture of NH$_2$NH$_2$/MeOH (7:3). (Glyco)peptides were then released from the resin, and all acid sensitive side-chain protecting groups were simultaneously removed using TFA 95%, TIS (triisopropylsilane) 2.5% and H$_2$O 2.5%, followed by precipitation with cold diethyl ether. The crude products were purified by HPLC on a Phenomenex Luna C18(2) column (10 µm, 250 mm × 21.2 mm) and a dual absorbance detector, with a flow rate of 10 mL/min.

**Peptide preparation.** All the peptides used in this work were dissolved at 100 mM in buffer 25 mM Tris-HCl pH 7.5. The pH of each solution was measured with pH strips and when needed adjusted to pH 7–8 through the addition of 0.1–5 µL of 2 M NaOH.

**NMR experiments.** All NMR experiments were recorded on a Bruker Avance III 600 MHz spectrometer equipped with a 5 mm inverse detection triple-resonance cryogenic probe head with z-gradients. The $^1$H-NMR resonances of the (glyco)peptides were completely assigned through standard 2D-TOCSY (30 and 80 ms mixing time), 2D-NOESY (400 ms mixing time) and 2D $^1$H,$^{13}$C-HSQC experiments at 283 and 298 K. The α-O-methyl-GalNAc (Carbosynth, MM06786) was assigned with 2D-TOCSY/NOESY and $^1$H,$^{13}$C-HSQC experiments at 298 K. Typical concentrations were around 1 mM for the heteronuclear experiments in 25 mM Tris(D$_{11}$)-DCl buffer, pD 7.5, with 150 mM of NaCl in H$_2$O/D$_2$O (90:10). The resonance of 2,2,3,3-tetradeutero-3-trimethylsilylpropionic acid (TSP) was used as a chemical shift reference in the $^1$H-NMR experiments ($\delta$ TSP = 0 ppm).

**STD NMR experiments.** STD NMR experiments were performed using a 1:35 molar ratio, defined by 20 µM *Dm*C1GalT1$^{T43-Q388}$ and 710 µM ligands (α-O-methyl-GalNAc, and (glyco)peptides), and 25 mM Tris(D$_{11}$)-DCl buffer, pD 7.5, with 150 mM NaCl and 150 mM MnCl$_2$ in D$_2$O. Some of the STD NMR experiments were accomplished in absence and presence of UDP (135 µM). In the presence of 150 µM MnCl$_2$, strong paramagnetic relaxation enhancements are observed for UDP, which prevents the observation of UDP proton signals in the NMR spectra. However, the presence of MnCl$_2$ does not preclude the observation of the proton signals of the ligands and allows to extract information from STD experiments. STD NMR spectra (stddiffesgp pulse sequence from Bruker pulse program library) were acquired with 1728 scans and 64 K data points, in a spectral window of 12335.53 Hz centered at 2818 Hz. Selective saturation (on resonance) was performed by irradiating at 7 and/or −0.5 ppm (depending if the ligand contains or not aromatic residues) using a series of 40 Eburp2.1000-shaped (from Bruker shaped pulses library) 90° pulses (50 ms) for a total saturation time of 2 s, and a relaxation delay of 4 s. For the reference spectrum (off resonance), the samples were irradiated at 100 ppm. Proper control experiments were performed for each ligand in the absence of protein and residual STD signals of the methyl groups of Ala/Thr were observed. This result was taken in account (subtracted) when analyzing the STD experiment in presence of *Dm*C1GalT1$^{T43-Q388}$. Protein control experiments were also accomplished using *Dm*C1GalT1$^{T43-Q388}$ in absence of a ligand and also subtracted from STD experiment. The STD spectrum ($I_{STD}$) was obtained by subtracting the on-resonance spectrum ($I_{on}$) to the off-resonance spectrum ($I_{off}$). The % of STD ($I_{STD}/I_{off} \times 100$) was estimated by comparing the intensity of the signals in the STD spectrum ($I_{STD}$) with the signal intensities of the reference spectrum ($I_{off}$). The STD amplification factor (STD$_{AF}$) was also estimated by multiplying the % STD values by the ligand excess[34], which in the case of our experiments was 35 for every ligand. To determine the STD-derived epitope map the relative % of STD were calculated by setting to 100% the STD signal of the proton with the highest STD and calculating the others accordingly (Supplementary Tables 2–4). Some protons were not able to be assessed with accuracy due to the use of water suppression or low signal/noise ratio and display a blue circle in the STD-derived epitope maps. Moreover, the resonances overlapped on the $^1$H-NMR spectrum were considered in STD estimation and are labelled as *.

**Molecular dynamics (MD) simulations.** The crystal structure of *Dm*C1GalT1-UDP-Mn$^{2+}$-APDT*RP was superimposed with the human B3GNT2-UDP-GlcNAc complex (PDB entry 7JHL), providing the coordinates of the UDP-GlcNAc in an identical location as that found for the UDP bound to *Dm*C1GalT1 (see Fig. 3e illustrating that B3GNT2 and *Dm*C1GalT ligands superimpose very well). Once we generated the *Dm*C1GalT1-UDP-GlcNAc-Mn$^{2+}$-APDT*RP complex, we replaced the UDP-GlcNAc by UDP-Gal resulting in the UDP-Gal-Mn$^{2+}$-APDT*RP complex. The other complexes were generated by mutating and adding or removing the corresponding residues with PyMOL 2.5. The calculations were carried out using AMBER 20 package, which was implemented with ff14SB and GLYCAM[65] force fields Each complex was immersed in a water box with a 10 Å buffer of TIP3P water molecules. The system was neutralized by adding explicit counter ions (Na$^+$ or Cl$^-$). A two-stage geometry optimization approach was performed. The first stage minimizes only the positions of solvent molecules, and the second stage is an unrestrained minimization of all the atoms in the simulation cell. The systems were then gently heated by incrementing the temperature from 0 to 300 K under a constant pressure of 1 atm and periodic boundary conditions. Harmonic restraints of 30 kcal mol$^{-1}$ were applied to the solute, and the Andersen

temperature-coupling scheme was used to control and equalize the temperature. The time step was kept at 1 fs during the heating stages, allowing potential inhomogeneities to self-adjust. Long-range electrostatic effects were modelled using the particle-mesh-Ewald method. An 8 Å cut-off was applied to Lennard-Jones interactions. Each system was equilibrated for 2 ns with a 2 fs time step at a constant volume and temperature of 300 K. Production trajectories were then run for additional 0.5 µs under the same simulation conditions. Adaptively Biased Molecular Dynamics method[66] implemented in AMBER 20 was used to calculate the free-energy maps for the APDT*RP and APDS*RP glycopeptides in water at 300 K.

**Cell culture**. All isogenic glycoengineered HEK293 cell lines were cultured in DMEM (Sigma-Aldrich) supplemented with 10% heat-inactivated fetal bovine serum (Sigma-Aldrich) and 2 mM GlutaMAX (Gibco) in a humidified incubator at 37 °C and 5% $CO_2$.

**CRISPR/Cas9-targeted KO in HEK293 cells**. CRISPR/Cas9 KO was performed using the GlycoCRISPR resource containing validated gRNAs libraries for targeting of all human GTs[67]. In brief, the previously developed HEK293core1 (KO *GCNT1/ ST3GAL1/2/ST6GALNAC2/3/4*) cells with stable-expression of secreted GFP-MUC1 reporter[30] were grown in sixwell plates (NUNC) to ~70% confluency and transfected with 1 µg of gRNA targeting *C1GALT1* gene (primer: GTAAAG-CAGGGCTACATGAG) and 1 µg of RFP-tagged Cas9-PBKS using lipofectamine 3000 (ThermoFisher Scientific) following the manufacturer's protocol. Twenty-four-hours post-transfection, cells were bulk-sorted with RFP expression by FACS sorter (SONY SH800). After 1 week of culture, the bulk-sorted cells were further single cell-sorted into 96-well plates. KO clones were screened by Indel Detection by Amplicon Analysis PCR with the primers (forward primer: 5'-CCTGCTG TGGGACTGAAAAC-3'; reverse primer: 5'-TGCATCTCCCCAGTGCTAAG-3') amplifying gRNA targeting sites and were further verified by Sanger sequencing.

**Construction of C1GalT1 enzyme and site-directed mutants**. The codon-optimized full coding human *C1GALT1* containing a C-terminal Myc-tag was synthesized by Genewiz USA and subcloned into EPB71 vector (Addgene ID 90018) for AAVS1 targeting KI. The site-directed mutagenesis was performed by Genscript with targeting the six candidate amino acid residues (R140, Y201, Y206, D240, W285, and Y310) replaced to Ala.

**ZFN-mediated KI of C1GalT1 variants in HEK293^Tn cells**. For site-directed knock-in (KI) a modified ObLiGaRe targeted AAVS1 safe harbor site KI strategy utilizing two inverted ZFN binding sites flanking the *C1GALT1* variants in donor plasmids were used[68]. KI was performed as described before for targeted KO with 1 µg of each ZFN tagged with GFP/Crimson and 2 µg donor plasmid. 48 h after transfection the 10–15% most highly expressed cell pool (KI pool) for both GFP and Crimson were enriched by FACS (SONY SH800). After 1 week of culture, the bulk-sorted cells were single cell-sorted into 96-well plates. The targeted KI single clones were screened by PCR using a primer pair specific for the junction area between the donor plasmid and the human AAVS1 locus, as well as a primer pair flanking the targeted KI locus. An allele-specific WT PCR (forward primer: 5'-CCTTACCTCTCTAGTCTGTGCTAG-3'; reverse primer: 5'- CGTAAGCAAA CCTTAGAGGTTCTGG-3') is used to verify the copy number of KI gene.

**Flow cytometry analysis**. The level of core 1 structure on cell surface was measured by flow cytometry with mouse mAb 3C9 (an in-house produced antibody) specific to core 1 glycosylation[69]. Cells were incubated on ice with 3C9 mAb (undiluted hybridoma supernatant which is equivalent to 1:1 dilution) for 30 min, followed by washing and incubation with Alexa Fluor 647 conjugated goat anti-mouse IgM (1 µg/mL) (Invitrogen, catalogue: A21235) for 30 min. Diluting and washing was performed in PBS with 1% BSA and cells were resuspended for flow cytometry analysis (SONY SA3800). Mean fluorescent intensity of the binding of mAb 3C9 populations was quantified by FlowJo software (FlowJo LLC).

**Immunocytology**. Cells were fixed with cold acetone for 10 min and incubated with anti Myc tag mAb 9E10 (undiluted hybridoma supernatant which is equivalent to 1:1 dilution) (ATCC, catalogue: CRL-1729) and mAb 3C9 (undiluted hybridoma supernatant which is equivalent to 1:1 dilution) overnight at 4 degree, followed by secondary Alexa Fluor 594 conjugated goat anti-mouse IgM (1 ug/mL) (Invitrogen, catalogue: A-21044). All samples were imaged using a Zeiss Axioskop 2 plus with an AxioCam MR3 followed by analysis with ImageJ (NIH).

**Reporting summary**. Further information on research design is available in the Nature Research Reporting Summary linked to this article.

## Data availability

The crystal structure of the *Dm*C1GalT1-UDP-APDT*RP complex was deposited at the RCSB PDB with accession code 7Q4I. Previously published PDB structures used in this

study are available under the accession codes: 7JHN, 7JHL, 6WMO, 2J0A, and 2J0B. The molecular dynamics simulations data have been deposited in the repository "open science framework" and can be found in the following link: "https://osf.io/sx2y4/?view_only= e68258f05a624223aeb987b630bd0f2a". Other data are available from the corresponding author upon reasonable request. Source data are provided with this paper.

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

## Acknowledgements

We thank the Diamond Light Source (Oxford, UK) synchrotron beamline I24 (experiment number MX20229-11). We thank ARAID, the Agencia Estatal de Investigación (AEI; BFU2016-75633-P and PID2019-105451GB-I00 to R.H.-G., and RTI2018-099592-B-C21 to F.C.), Gobierno de Aragón (E34_R17 and LMP58_18 to R.H-G.) with FEDER (2014–2020) funds for "Building Europe from Aragón" for financial support, and the Danish National Research Foundation (DNRF107). F.M., A.S.G. and H.Co. thank to Fundação para a Ciência e a Tecnologia for funding projects: IF/00780/2015; PTDC/BIA-MIB/31028/2017, UCIBIO project (UIDP/04378/2020 and UIDB/04378/2020) and i4HB project (LA/P/0140/2020). A.S.G. also acknowledges the PhD fellowship (SFRH/BD/140394/2018), and F.M. and H.Co. also thank the CEEC contracts (2020.00233.CEE-CIND and 2020.03261.CEECIND, respectively). The NMR spectrometers are part of the National NMR Facility supported by FCT-Portugal (ROTEIRO/0031/2013–PINFRA/22161/2016, co-financed by FEDER through COMPETE 2020, POCI and PORL and FCT through PIDDAC). A.M.G-R. thanks the Spanish Ministry of Science, Innovation and Universities for the FPI fellowship. The research leading to these results has also received funding from the FP7 (2007-2013) under BioStruct-X (grant agreement N°283570 and BIOSTRUCTX_5186).

## Author contributions

R.H.-G. designed the crystallization construct and solved the crystal structure. A.M.G.-R. performed the expression and purification of all proteins, the enzyme kinetics and ITC experiments, and crystallized the complex. A.M.G.-R. refined the crystal structure. F.C. performed the molecular mechanics and MD calculations. I.C. synthetized the glycopeptides. A.S.-G. and H.Co. performed the STD NMR experiments. Z.Y. and Y.N. performed the *in cells* activity of C1GalT1 mutants. R.H.-G. wrote the article with mainly the

contribution of F.M., F.C., Y.N., A.M.G.-R., and H.C. All authors read and approved the final paper.

## Competing interests

The authors declare no competing interests.
