## [Peer Review File · Nature Communications]

nature portfolio

Peer Review File

Draft OnlyREVIEWER COMMENTS

Reviewer #1 (Remarks to the Author):

This manuscript by Gonzales-Ramirez et al focuses on the important glycosyltransferase C1GalT1. The manuscript is overall extremely well-written and the logical progression of experiments present a comprehensive story that reveals a mechanistic understanding of the enzyme. The structural studies taken together with other results reveal important findings including a putative role for COSMC in dimerization, an explanation of the lack of peptide specificity near the O-GalNAc site for Gal addition, the somewhat surprising finding of not using an induced fit mechanism, and the rationale for STn (that along with Tn are cancer biomarkers) not being a substrate for the enzyme (due to multiple steric clashes). Overall enthusiasm for this manuscript is extremely high though there are a few very minor details listed below.

1. line 64 replace "extend" with "extent"
2. line 84, define "it"
3. kcat in all cases is very low with somewhat high Km values, did the authors attempt to use molecular crowders to improve kinetics?
4. line 159 reword "worse recognized"
5. Figure 4D was washed out and lacked resolution (though readable) on reviewers PDF
6. Is there any additional structural evidence that can be added and discussed in the discussion regarding other "2nd step in the pathway enzymes" using the high-energy confirmation proposed mechanism?

Reviewer #2 (Remarks to the Author):

The manuscript describes new scientific findings, i.e., the structure of an inverting glycosyltransferase, C1GalT1, which synthesizes the so-called core 1 structure, and specific interactions of this enzyme with donor and acceptor substrates. Core 1 disaccharide linked to Ser or Thr is a precursor for mucin-type O-glycans, and impaired biosynthesis is associated with different diseases, including cancer. Understanding the complexity of the underlying glycosylation machinery will be important for the development of therapeutic approaches. The study employs several biophysical techniques, ranging from X-ray crystallography to NMR to isothermal titration

calorimetry (ITC), and is complemented by molecular dynamics simulations as well as in vitro (in cells) activity assays of wild type and mutant enzyme. The findings are important for the field of glycobiology and will also shine new light on more general cell-biology associated questions. Therefore, I support publication of the results in Nature Communications.

The paper is well structured but challenging to follow in parts. Especially, for colleagues not from the field it may be difficult to fully appreciate the importance of some of the findings. Some suggestions how to improve this are included with this report. Clearly, the paper would benefit from another round of critical proof reading by the authors.

The experimental data are sound and in general justify the conclusions. There are some issues concerning the NMR part that need to be addressed, as this is detailed below. The structural data file 7Q4I is not available yet from the RCSB. Would the data set be available for review purposes?

General comments:

1/ For the non-glycobiologist a scheme reflecting the biosynthesis of core 1 and core 3 structures would be helpful.

2/ The main structural features are based on crystal structure analysis. STD NMR has been extensively used to complement these data with solution-structure data. As far as I can see NMR data and crystallographic data correspond well. It may be valuable to have a short summary or table, e.g., at the end of the discussion part, addressing the question where these data are mutually reinforcing or complementing.

The following detailed comments follow the manuscript and are not ranked by importance.

Detailed comments:

p. 2, l. 30: Why is it "optimal binding" and not just "binding"?

p. 2, l. 36: I would substitute "sites" with "types" since "site" may be misleadingly associated with a site on the enzyme.

p. 7, l. 140: STD NMR is a specialized NMR technique to study ligand binding and may not be so present to the non-NMR specialist. Therefore, at least a reference to the original publications (Mayer & Meyer, 1999 and Mayer & Meyer, 2001) is required. A few sentences describing what kind of information is derived from such experiments would help. In the context of this study it would be important to briefly introduce the concept of STD NMR based ligand-epitope mapping.

p. 7, l. 143 - 145: Instead of "saturation frequency" the term "on-resonance frequency" should be used. "Performed" and "considered" should be substituted with, e.g., "used". The sentence should be split in two. Suggestion: "... region. For P4 only the aliphatic on-resonance frequency was used."

p- 7, l. 146: I am surprised there are no line broadening effects observed due to the presence of Mn²⁺ ions. Other NMR studies on glycosyltransferases have substituted Mn²⁺ with Mg²⁺, which reduces enzymatic activity but prevents effects from paramagnetic line broadening. To my experience, even traces of paramagnetic ions are rather detrimental to spectra quality. The authors need to comment on this.

p. 7, l. 157: What kind of "amplification factor" has been used? In the experimental part, p. 28/29, l. 674 - 676 it is stated that "STD values were estimated by comparing the intensity of the signals in the STD spectrum with the signal intensities of the reference spectrum". It needs to be specified exactly which formula has been used to calculate the STD amplification factor. The original definition of the STD amplification factor is the fractional saturation of a proton multiplied by the excess of ligand over protein (cf. Mayer & Meyer, 2001). This definition makes the STD amplification factor independent of specific sample conditions, i.e., varying ligand and protein concentrations.

p. 7, l. 158: " α -methyl-GalNAc" should be substituted with " α -O-methyl-GalNAc".

p. 7, l. 157 and following: From lower STD amplification factors (see above) it is concluded that binding of the methyl glycoside of GalNAc is weaker ("worse") than for the glycopeptide APDT*RP. This conclusion is not strictly correct. Binding affinity and the size of STD amplification factors are not necessarily correlated. To come to such a conclusion a titration or competition experiment would be required. Please phrase this more carefully, e.g., "the lower STD amplification factors may reflect the expected lower binding affinity of GalNAc vs. ...".

p. 8, l. 168: A "moiety" cannot be a "driving force". It would be more correct to say "... binding of the GalNAc moiety is the driving force ..."

p. 8, l. 179 and following, and p. 9, l. 193 - 196: "Then, we evaluated whether this enzyme requires UDP binding prior to binding the glycopeptides". I cannot see any arguments that allow to discriminate between an ordered or a random sequential, or other possible mechanisms?

p. 13, l. 291: Here, and in some other instances the "l" in C1GalT1 has been lost.

p. 16, l. 382 - 383, and p. 17 l. 405: How exactly are the structures consistent with the STD effects? It would help to have a link to a certain spectral feature. The figures referred to only show structures.

p. 18, l. 425: I am not sure I understand why this explains that exposed Tn epitopes are only found in cancer cells.

p.20: I am struggling with the last sentence, which is rather complicated. I would prefer a clear message at the end.

p. 25, ITC experiments: All the curves (thermograms) are missing the "tail". Please discuss this with respect to the data reliability.

p. 28, l. 661: 150 μ M MnCl₂ is not consistent with the concentration given in the figure legends.

p. 28, l. 665 - 667: The pulse cascade nomenclature "Eburp2-1000" and the name of the pulse program "stddiffesgp" is Bruker nomenclature. It should be stated that this has been taken from the Bruker pulse program library.

p. 35, l. 839 and following (legend Fig. 4): The dihedral angles used need to be defined explicitly in the legend.

Supplementary Information:

Fig. S4: 1/ The difference spectra and the reference spectra are shifted. Why?

2/ I would say both STD spectra show a "response". It is weak but it is not that there is "no response".

3/ Assignment of signals?

4/ Why is the region showing the UDP signals not shown? Regarding the presence of Mn^{2+} , it would be interesting to search for specific line-broadening effects on UDP-signals. Can this be included?

5/ Minor: The phrasing "different irradiations" should be substituted with "different on-resonance frequencies".

Fig. S5: Impurities needs to be marked in all spectra. The H6 protons show rather intense signals. Are the integrals OK? The resolution (pixel quality) of the spectra is rather low.

Fig. S6: It would be nice to have the parameters ΔG , ΔH , etc. as an insert.

Tables S2-S4: Definition of STD_{AF}.

Reviewer #3 (Remarks to the Author):

The rationale of the work is that studies of rat, human and fly C1GalT1 orthologues suggest that the enzymatic activity is affected by the protein sequence. In the absence of a structure of C1GalT1 with bound substrates, the molecular basis for peptide sequence recognition is not known. The key result of the work is the determination of the C1GalT1 structure with bound substrate (peptide + sugar), which shows a strained conformation for the peptide, and allows a description of the catalytic mechanism to be devised.

The authors link their study to the tumour-associated T and Tn antigens. The importance of their work to cancers is tenuous at best without a clear rationale provided in the introduction. Actually, they don't consider the implications of their work to tumours in any detail in their manuscript at all. I suggest removing these connections, especially in title, as it does not reflect the work.

Enzyme kinetic studies. The data does not support the claim that "...peptide sequence is crucial for achieving optimal kinetic parameters..." Firstly, the difference in the parameters is not particularly large and probably within error. Secondly, the kinetic parameters determined are 'apparent' kinetic constants (and should be labelled as such), since the catalysed reaction is a two substrate reaction and they have kept the concentration of UDP-constant at a sub-saturating concentration. The authors repeat the kinetic experiments varying also the UDP-Gal concentration and fit to ping-pong

or ternary complex kinetic models. It is likely the different peptides could subtly alter the K_m for UDP-Gal, and this could explain the perceived differences in the apparent kinetic parameters. The finding that APDT*RP only binds in the presence of UDP (Fig2b) is very interesting and suggests that the kinetic mechanism is altered (ordered binding vs. random binding)—this also argues that the authors should determine the kinetics varying both substrates. Please add error bars to the k_{cat}/K_m plot in Figure 1c.

NMR experiments. This reviewer is not an expert in NMR. The key finding here is that it is the sugar that drives binding, with only minor contributions by the peptide.

Allostery. The enzyme DmC1GalT1T43-Q388 does appear to have a sigmoidal curve (supplementary figure7 purple data). The authors should fit this data to a hill model and compare this to a standard Michaelis-Menten model to determine whether for APDT*RP binding is allosteric. Also, please elaborate on why your results “...imply that DmC1GalT1 does not likely follow an induced-fit mechanism...”

The structural analysis is solid, although there are already structures of homologues in the pdb, which perhaps renders the work incremental. The key finding being that the substrate is held in the active site in a strained conformation, but that the substrates are well placed for an inversion of configuration SN2 mechanism. The residues important for substrate binding and catalysis are further probed using mutagenesis (in vitro and in cell, which is a very nice assay).

The MD simulations help explain the different binding patterns seen for the peptides, but otherwise don't add significantly to the work without accompanying experimental data to backup these models of the apo and UDP-Gal structures.

Other comments.

1) for the statement “Nevertheless, it is predicted that C1GalT1 efficiently transfers Gal to all GalNAc moieties (Tn) on proteins indiscriminately and independently of the underlying peptide sequences and clustering of GalNAc O-glycans” please provide a reference. Or, explain why you predict this.

2) In figure 3b, it is not clear what this density is. As stated it is difference density? Usually difference density is red (-3 sigma) and green (+3 sigma). Perhaps it is from an omit experiment?

3) The link to Cosmc driving oligomerisation is not substantiated by the data and should be removed. (lines 221-227)

4) Figure 4, a ligPlot would aid the reader.

5) The introduction should provide more foundational/basic information to help orient the reader and to broaden the readership—for example a figure showing T and Tn antigen synthesis would help.

6) Figure 4e, the mechanism is missing an arrow, a charge is incorrect, and the transition state has a covalent bond that should be a transitional/breaking bond.

Editorials:

Line 64 extend should be extent?

Line 299 present should be predicted

Line 337 Arg125 should be Arg152?

Figure 3c – what does the orange colour denote?

Reviewer #4 (Remarks to the Author):

The manuscript characterizes the interaction between various glycopeptides with the galactose transferase C1GalT1. Apart from kinetic, NMR, ITC and crystallographic studies, the experiments are complemented by molecular dynamics simulations of various complexes. I will focus my review on the molecular simulations. One of the key findings from the simulations is the observation that for the free glycopeptide, the Psi angle of the glycosidic bond is preferably found at an angle of 120 degree, while in the crystal structure it is seen at 180 degree. This is an interesting finding, corresponding to a conformational selection model and possibly explaining why the enzyme is also active on Ser-glycosylated peptides, in which the 180 degree angle is preferred also for the free peptide. The work seems carefully done and reads well.

I would like to make the following comments:

1. The observed conformational preferences for the Thr-peptide, could be interpreted by a conformational selection model on the substrate. In line 311, the authors refer to the conformational penalty as an entropic penalty, which is apparently overcome by favorable interactions. I do not quite agree that this effect is purely entropic; energetic contributions will also contribute to the preference of the free peptide. In fact, the authors should be able to quantify the free-energy penalty of bringing Psi from 120 to 180 degree by direct counting the conformation with

angles > 150 and < 150 degree. From [doi: 10.1002/cphc.201900079], I estimate this value to be about 10 kJ/mol. A proper integration over the conformational regions of interest as in [doi: 10.1021/acs.jcim.7b00351] could possibly reduce this value. The authors should attempt to quantify the preferences from the simulations of the free peptide.

2. As the APDS*RP peptide shows slightly worse affinity, and this peptide will not have the conformational penalty mentioned in the previous point, the conformational penalty of APDT*RP should be compensated by the transfer of the hydrophobic CH₃ group in Thr from water to a favorable CH- π interaction. The authors could emphasize that rather subtle free-energetic effects probably guide the binding.

3. I would encourage the authors to provide a table with a clear overview of the interactions determined by the STD NMR experiments and the observations in the MD simulations. Are all the relevant atoms indeed in close proximity?

4. Can the authors say more about the failed crystallography experiments with the other glycopeptides (page 9)? This seems to be in contrast to the tighter binding and higher activities for these peptides?

5. Some minor textual glitches:

a. Line 64: extend -> extent

b. Line 140: We performed then -> We then performed

c. Line 191: these somewhat were complex -> these were somewhat complex

d. Lines 291, 298: DmC1GaT1 -> DmC1GalT1

e. Lines 326 to 328: The activity ... was completely inactive -> Rephrase

REVIEWER

COMMENTS

Reviewer #1 (Remarks to the Author):
This manuscript by Gonzales-Ramirez et al focuses on the important glycosyltransferase C1GalT1. The manuscript is overall extremely well-written and the logical progression of experiments present a comprehensive story that reveals a mechanistic understanding of the enzyme. The structural studies taken together with other results reveal important findings including a putative role for COSMC in dimerization, an explanation of the lack of peptide specificity near the O-GalNAc site for Gal addition, the somewhat surprising finding of not using an induced fit mechanism, and the rationale for STn (that along with Tn are cancer biomarkers) not being a substrate for the enzyme (due to multiple steric clashes). Overall enthusiasm for this manuscript is extremely high though there are a few very minor details listed below.

Thank you for your comments.

1. line 64 replace "extend" with "extent"

Response#1: Thank you for pointing this out.

Action#1: The spelling has been corrected.

2. line 84, define "it"

Response#2: "it" refers to C1GalT1.

Action#2: "it" has been replaced by the enzyme

3. kcat in all cases is very low with somewhat high Km values, did the authors attempt to use molecular crowders to improve kinetics?

Response#3: Yes, we used BSA, a typical crowder used in enzyme kinetic assays. However, we forgot to mention this in the methodology. Note that we performed the reactions with and without BSA and found that BSA improved significantly the activity. In our assay, BSA improved *DmC1GalT1* activity 2-3-fold with respect to the enzyme activity under the same conditions without BSA. In addition, we do not know why *DmC1GalT1* is slow but we used fresh *DmC1GalT1* in all our assays. In the past, we found other proteins such as FUT8 to be a slow enzyme. Both FUT8 and C1GalT1 glycosylate hundreds of acceptor sites, yet, they are slow enzymes in vitro under our conditions. In the laboratory, we also work with other enzymes such as NleB1, a promiscuous and multi-substrate enzyme, that is much faster than FUT8 or *DmC1GalT1*. Overall, we do not know why enzymes such as *DmC1GalT1* or FUT8 are so slow in catalysis. Perhaps these enzymes are faster within the Golgi environment.

Action#3: The concentration of BSA has been now indicated in Methods.

4. line 159 reword "worse recognized"

Response#4: Thank you for pointing this out.

Action#4: Now, it states as "recognized worse than..."

5. Figure 4D was washed out and lacked resolution (though readable) on reviewers PDF

Response#5: Thank you for pointing this out.

Action#5: Fig. 4E (before Figure 4D) in the revised version of our manuscript has been updated to improve its resolution.

6. Is there any additional structural evidence that can be added and discussed in the discussion regarding other "2nd step in the pathway enzymes" using the high-energy confirmation proposed mechanism?

Response#6: Thank you for your comment. We have been thinking more about your comment. We think that by applying this strategy, the enzyme can recognize both acceptor substrates in a similar way without significantly changing its active site. Otherwise, if C1GalT1 did not impose the staggered conformation to the α -GalNAc-Thr, the enzyme active site would need to have more plasticity, which appears not to be the case.

Action#6: We have added the following text in the Discussion section, see below:

"It is tempting to speculate that it might be likely more structurally and energetically advantageous for C1GalT1 to impose the staggered conformation to α -GalNAc-Thr, which is low populated in solution, than to adapt the active site to the main conformer found in solution for α -GalNAc-Ser (staggered conformation) and α -GalNAc-Thr (eclipsed conformation)".

Reviewer #2 (Remarks to the Author):

The manuscript describes new scientific findings, i.e., the structure of an inverting glycosyltransferase, C1GalT1, which synthesizes the so-called core 1 structure, and specific interactions of this enzyme with donor and acceptor substrates. Core 1 disaccharide linked to Ser or Thr is a precursor for mucin-type O-glycans, and impaired biosynthesis is associated with different diseases, including cancer. Understanding the complexity of the underlying glycosylation machinery will be important for the development of therapeutic approaches. The study employs several biophysical techniques, ranging from X-ray crystallography to NMR to isothermal titration calorimetry (ITC), and is complemented by molecular dynamics simulations as well as in vitro (in cells) activity assays of wild type and mutant enzyme. The findings are important for the field of glycobiology and will also shine new light on more general cell-biology associated questions. Therefore, I support publication of the results in Nature Communications. The paper is well structured but challenging to follow in parts. Especially, for colleagues not from the field it may be difficult to fully appreciate the importance of some of the findings. Some suggestions how to improve this are included with this report. Clearly, the paper would benefit from another round of critical proof reading by the authors. The experimental data are sound and in general justify the conclusions. There are some issues concerning the NMR part that need to be addressed, as this is detailed below. The structural data file 7Q4I is not available yet from the RCSB. Would the data set be available for review purposes?

Response#: Thank you for your comments. We have proof read and improved our article and we think now that our manuscript is now more understandable for colleagues not coming from the Glycobiology field.

Action#: We have uploaded the pdb and mtz file in the NCOMMS tracking system in case you want to double check our structure.

General

comments:

1/ For the non-glycobiologist a scheme reflecting the biosynthesis of core 1 and core 3 structures would be helpful.

Response#1: The reviewer is right and a scheme would help the reader to understand these types of molecules.

Action#1: We have included a scheme in Fig. 5a that shows the biosynthesis of core 1, core 2, core 3, STn and complex O-glycans. The corresponding figure legend describes the content of that particular scheme.

2/ The main structural features are based on crystal structure analysis. STD NMR has been extensively used to complement these data with solution-structure data. As far as I can see NMR data and crystallographic data correspond well. It may be valuable to have a short summary or table, e.g., at the end of the discussion part, addressing the question where these data are mutually reinforcing or complementing.

Response#2: The reviewer is right on the short summary or table in the discussion part.

Action#2: We have included the following text in the discussion section, see below,

“Herein, the integration of X-ray crystallographic data and STD NMR allowed to decode the recognition of APDT*RP by *DmC1GalT1*. From the visual inspection of the complex’s crystal structure the APDTR*RP is mainly recognized through GalNAc unit by a network of H-bonds involving OH3, OH4 and OH6. This observation is complemented by the STD NMR spectra of APDT*RP in presence of *DmC1GalT1*, which provides information about GalNAc aliphatic protons, and pinpoints that GalNAc protons are those in closer contact with the protein, reinforcing the conclusion that GalNAc is the main contact point to the enzyme. With respect to the peptide both techniques indicate that Pro residues and the methyl group of Thr are involved in the recognition, helping to stabilize the peptide by a mix of hydrophobic and hydrogen bond interactions”.

”.

The following detailed comments follow the manuscript and are not ranked by importance.

Detailed comments:

p. 2, l. 30: Why is it "optimal binding" and not just "binding"?

Response#: The reviewer is right and we do not need the word “optimal”.

Action#: We have removed the word “optimal” and left only “binding”.

p. 2, l. 36: I would substitute "sites" with "types" since "site" may be misleadingly associated with a site on the enzyme.

Response#: We think that it is convenient to remove “sites” to better understand the sentence.

Action#: We have removed the word “sites”.

p. 7, l. 140: STD NMR is a specialized NMR technique to study ligand binding and may not be so present to the non-NMR specialist. Therefore, at least a reference to the original publications (Mayer & Meyer, 1999 and Mayer & Meyer, 2001) is required. A few

sentences describing what kind of information is derived from such experiments would help. In the context of this study it would be important to briefly introduce the concept of STD NMR based ligand-epitope mapping.

Response#: We agree with the reviewer and think that adding a brief information on STD or the ligand-epitope mapping would be useful for the readers.

Action#: We have added the following text in the manuscript, see below,
“The STD NMR experiment is a ligand-observed technique (only the ¹H-NMR assignment of the ligand is required for analysis) that relies on saturation transfer, through nuclear Overhauser effect, from receptor (e.g. protein/enzyme) proton resonances to protons of a ligand (e.g. carbohydrate, glycopeptide) exchanging between a protein-bound and free state³³. Analysis of the STD responses allows to infer which atoms of the binding ligand are in closer contact with the receptor, and to determine the so-called STD-derived epitope mapping³⁴”.

p. 7, l. 143 - 145: Instead of "saturation frequency" the term "on-resonance frequency" should be used. "Performed" and "considered" should be substituted with, e.g., "used". The sentence should be split in two. Suggestion: "... region. For P4 only the aliphatic on-resonance frequency was used."

Response#: We agree with the reviewer in the suggested changes.

Action#: All changes suggested by the reviewer have been addressed and now we think that the text reads better.

Draft Only

p- 7, l. 146: I am surprised there are no line broadening effects observed due to the presence of Mn^{2+} ions. Other NMR studies on glycosyltransferases have substituted Mn^{2+} with Mg^{2+} , which reduces enzymatic activity but prevents effects from paramagnetic line broadening. To my experience, even traces of paramagnetic ions are rather detrimental to spectra quality. The authors need to comment on this.

Response#: We agree with the reviewer comment.

We were aware of the possible deleterious effect of the paramagnetic relaxation enhancement (PRE) effect on the spectral line widths due to the paramagnetic Mn^{2+} ion. However, the PRE effect depends on the distance of the observed nucleus to the paramagnetic ion and in solution or in the protein-bound state the UDP molecule is directly complexed to Mn^{2+} through the phosphate chain preventing the observation of UDP signals. Fortunately, this was not the case of the glycopeptide, for which sharp resonances are observed in the presence of $MnCl_2$ (150 μM) either in absence or presence of the protein (see below Figure 1B). For the STD experiments, a concentration of 150 μM of $MnCl_2$ was used and molar ratios of $\sim 0.9:1$ UDP/ $MnCl_2$ and $\sim 5:1$ glycopeptide/ $MnCl_2$ were employed. In these conditions, strong PRE effect is observed for UDP signals (Figure 1A) but not for the glycopeptide (Figure 1B).

Figure 1 A. 1H -NMR UDP vs 1H -NMR UDP + $MnCl_2$ vs 1H -NMR UDP+ $MnCl_2$ + C1GalT1 B. 1H -NMR PEP 4 vs 1H -NMR PEP4 + $MnCl_2$ vs 1H -NMR PEP4 + $MnCl_2$ + C1GalT1

STD experiments of APDT*RP were also carried out in absence of $MnCl_2$ (see below Figure 2A) and in the presence of 5 mM $MgCl_2$ (Figure 2B). However, only in the presence of Mn^{2+} (Figure 2C) STD response in the glycopeptide proton signals was observed (particularly evident at the GalNAc proton region). These experiments pinpoint the requirement of Mn^{2+} for specific binding and that only in these conditions it can be extracted some binding information from STD based methods.

Figure 2. **STD experiments of APDT*RP at 710 μ M in presence of 20 μ M C1GalT1 and 135 μ M UDP.** A. Off resonance (grey) and STD spectra (black) in absence of MnCl_2 . B. Off resonance (grey) and STD spectra (black) in presence of 5 mM MgCl_2 . C. Off resonance (grey) and STD spectra (black) in presence of 150 μ M MnCl_2 .

Action#: We share this data with the reviewer because we believe that they are pertinent. In the manuscript we have added the following note in the experimental section, see below,

“In the presence of 150 μ M of MnCl_2 strong paramagnetic relaxation enhancements are observed for UDP, which prevents the observation of UDP proton signals in the NMR spectra. However, the presence of MnCl_2 does not preclude the observation of the proton signals of the glycopeptides and allows to extract information from STD experiments.”

p. 7, l. 157: What kind of "amplification factor" has been used? In the experimental part, p. 28/29, l. 674 - 676 it is stated that "STD values were estimated by comparing the intensity of the signals in the STD spectrum with the signal intensities of the reference spectrum". It needs to be specified exactly which formula has been used to calculate the STD amplification factor. The original definition of the STD amplification factor is the fractional saturation of a proton multiplied by the excess of ligand over protein (cf. Mayer & Meyer, 2001). This definition makes the STD amplification factor independent of specific sample conditions, i.e., varying ligand and protein concentrations.

Response#: We agree with the reviewer, and we clarify in the experimental section how the STD values were estimated and how the STD amplification factor was calculated. The STD amplification factor was calculated as the original formula in Mayer & Meyer, *J. Am. Chem. Soc.* **123**, 6108-6117 (2001). (reference 34 in the actual version of the manuscript).

Action#: We added the following text in the experimental section of the manuscript.

"The STD spectrum (I_{STD}) was obtained by subtracting the on-resonance spectrum (I_{on}) to the off-resonance spectrum (I_{off}). The STD values (I_{STD}/I_{off}) were estimated by comparing the intensity of the signals in the STD spectrum (I_{STD}) with the signal intensities of the reference spectrum (I_{off}). The STD amplification factor (STD_{AF}) was also estimated by multiplying the STD values by the ligand excess ($I_{STD}/I_{off} \times \text{ligand excess}$), which in the case of our experiments was 35 for every ligand."

p. 7, l. 158: " α -methyl-GalNAc" should be substituted with " α -O-methyl-GalNAc".

Action#: This has been fixed in the revised version of the manuscript.

p. 7, l. 157 and following: From lower STD amplification factors (see above) it is concluded that binding of the methyl glycoside of GalNAc is weaker ("worse") than for the glycopeptide APDT*RP. This conclusion is not strictly correct. Binding affinity and the size of STD amplification factors are not necessarily correlated. To come to such a conclusion a titration or competition experiment would be required. Please phrase this more carefully, e.g., "the lower STD amplification factors may reflect the expected lower binding affinity of GalNAc vs. ...".

Response#: We agree with the reviewer's comment.

Action#: In the revised version of the manuscript, the text has been changed according to the reviewer's comment as stated below,

"However, the STD amplification factor was lower in the case of α -O-methyl-GalNAc than that of the APDT*RP or P4 (Table 2, 3 and 4), which likely reflects the expected lower binding affinity of α -O-methyl-GalNAc versus those of the glycopeptides, inferred from the kinetics experiments".

p. 8, l. 168: A "moiety" cannot be a "driving force". It would be more correct to say "... binding of the GalNAc moiety is the driving force ..."

Action#: This has been fixed in the revised version of the manuscript.

p. 8, l. 179 and following, and p. 9, l. 193 - 196: "Then, we evaluated whether this enzyme requires UDP binding prior to binding the glycopeptides". I cannot see any arguments that allow to discriminate between an ordered or a random sequential, or other possible mechanisms?

Response#: We agree with the reviewer and in fact we do not mention anything regarding the potential kinetic mechanism in the manuscript because this is out of the scope of our article. In any case, we have clarified in the manuscript why we performed this kind of experiments with *DmC1GalT1*.

Action#: For clarification purposes, we have added the following text in the manuscript in the STD NMR section. It reads as follows,

"We conducted these experiments in the presence of absence of the nucleotide because previously we found that other distant GT-A fold GTs such as GalNAc-T2, and the NleB/SseK effector proteins were dependent on the presence of UDP for binding to their protein/peptide acceptor substrates, implying the existence of an induced-fit mechanism^{17,30,31}. Interestingly, in the case of GalNAc-T2, the active conformation of GalNAc-T2, characterized by the shifting of a flexible loop from an open to a closed conformation, was completely achieved in the presence of UDP-GalNAc and less in the presence of UDP¹⁷. We also found that the GT-B fold FUT8 showed similar properties to the other two GTs, though FUT8 bound better to an N-glycan in the presence of GDP, and the nucleotide was not essential for binding to the N-glycan³². In addition, NleB/SseK and FUT8 also contained flexible loops that were ordered in the presence of the nucleotide as we found for GalNAc-T2, implying that the active site adopted an active conformation once that these flexible loops bound to the sugar nucleotide^{17,30,31,33}. Overall, we proposed for these three enzymes that the binding of the sugar nucleotide was required for binding to the acceptor substrate (optimal binding for FUT8 in the presence of GDP) and in turn for glycosylation".

p. 13, l. 291: Here, and in some other instances the "l" in C1GalT1 has been lost.

Action#: This has been fixed in the revised version of the manuscript.

p. 16, l. 382 - 383, and p. 17 l. 405: How exactly are the structures consistent with the STD effects? It would help to have a link to a certain spectral feature. The figures referred to only show structures.

Action#: This has been fixed in the revised version of the manuscript and now Fig. 2a is linked to the reviewer's comment above.

p. 18, l. 425: I am not sure I understand why this explains that exposed Tn epitopes are only found in cancer cells.

Response#: We agree with the reviewer that that particular sentence is not clear enough.

Action#: To clarify that sentence, we have changed the previous text to the following,

"This suggests that the C1GalT1 can serve widely in core1 O-glycan elongation and cover the entire spectrum of O-glycans distributed in the proteome. Clearly, C1GalT1 may have different kinetic properties for GalNAc glycosylated O-glycopeptides, but in normal cells, most if not all O-glycans are elongated to mask exposure of the cancer-associated Tn structure. Exposure of Tn in cancer cells is generally not due to inactivating mutations in the *C1GalT1* gene¹⁸ and heterogeneous with both Tn and core 1 structure are found in most cancer cells⁵². Thus, reduced expression of C1GalT1 may instead lead to incomplete

O-glycan elongation with preferences for O-glycan sites that are less preferred substrates for C1GalT1”.

p.20: I am struggling with the last sentence, which is rather complicated. I would prefer a clear message at the end.

Action#: To facilitate the reading of this part of the article, we have made the following changes at the end of the Discussion section, see below,

“We also uncover that C1GalT1 imposes a high-energy and unfavorable conformation to α GalNAc-Thr as a required step for glycosylation. This is a remarkable example of how GTs have implemented strategies to promote conformational changes in the acceptor substrates to achieve glycosylation”.

p. 25, ITC experiments: All the curves (thermograms) are missing the "tail". Please discuss this with respect to the data reliability.

Response#: The reviewer is right on this matter. However, and in our case, our ITCs thermograms are low c-parameter systems ($c < 1$), explaining why the initial plateau is missing. Despite this, data reliability is still valid as has been previously described in the literature (see PMID: 17920027 and 14640663). The typical full sigmoidal curve could be theoretically obtained by increasing cell concentration in order to rise the c-parameter value. Nevertheless, this could not be achieved due to technical difficulties, primarily protein aggregation, but also peptide solubility. Despite this, it has been previously discussed that the estimated K_d in low c-parameter systems is still reliable as the possible error in n estimation has almost no effect in the K_a parameter fitting (see PMID: 17920027 and 14640663).

In addition, we have also added the correlation coefficients from our ITC fitting curve, which indicate that our fittings are of high quality and in turn the data are reliable.

Action#: We have added the following information in the legend in Figure 2, see below.

“All our ITCs thermograms have low Wiseman “c”-parameters ($c < 1$), explaining why the initial plateau is missing. Despite this, data reliability is still valid as previously described before^{62,63}, and in particular the estimated K_d s from the low Wiseman “c”-parameters are still reliable as the possible error in n estimation has almost no effect in the K_a parameter fitting. Note that the Wiseman “c”-parameter is the product of the receptor concentration and the association constant, K_a ⁶³.”. In addition, we have also added the references with PMIDs 17920027 and 14640663.

We have also added the correlation coefficients from our ITC fitting curves in Supplementary Table 5. The text is as follow,

“All the correlation coefficients (R^2) of the ITCs fittings were higher than 0.99”.

p. 28, l. 661: 150 μ M MnCl₂ is not consistent with the concentration given in the figure legends.

Response#: We have checked the concentration of MnCl₂ in the figure legends and we have not found any inconsistency with the concentration used for the STD NMR experiments. In all figure legends in SI, the concentration of MnCl₂ was 150 μ M.

Action#: No action taken.

p. 28, l 665 - 667: The pulse cascade nomenclature "Eburp2-1000" and the name of the pulse program "stddiffesgp" is Bruker nomenclature. It should be stated that this has been taken from the Bruker pulse program library.

Response#: The reviewer is right.

Action#: The comments by the reviewer has been added in the Methods section, see below,

“However, the presence of MnCl_2 does not preclude the observation of the proton signals of the ligands and allows to extract information from STD experiments. STD NMR spectra (stddiffesgp pulse sequence from Bruker pulse program library) were acquired with 1728 scans and 64 K data points, in a spectral window of 12335.53 Hz centered at 2818 Hz”.

and,

“Selective saturation (on resonance) was performed by irradiating at 7 and/or -0.5 ppm (depending if the ligand contains or not aromatic residues) using a series of 40 Eburp2.1000-shaped (from Bruker shaped pulses library) 90° pulses (50 ms) for a total saturation time of 2s, and a relaxation delay of 4s.”

p. 35, l 839 and following (legend Fig. 4): The dihedral angles used need to be defined explicitly in the legend.

Action#: We have added the dihedral angles as requested in the legend of Fig. 4, see below,

“The dihedral angles are defined as follow, $\phi = \text{O5-C1-O1-C}\beta$, and $\psi = \text{C1-O1-C}\beta\text{-C}\alpha$ ”.

Supplementary

Information:

Fig. S4: 1/ The difference spectra and the reference spectra are shifted. Why?
2/ I would say both STD spectra show a "response". It is weak but it is not that there is "no response".

Response#: Thank you for pointing this out. In the figure S4, one spectrum was calibrated, while the others were not. In the revised version we corrected this issue. In addition, while, revising the manuscript we realized that the spectra in figure S4 did not account the STD control of the peptide, which was also corrected in the new figure S4.

Action#: A new Fig. S4 was added to the revised manuscript.

3/ Assignment of signals?

Response#: Yes, thanks for the suggestion.

Action#: The assignments were added in all the Figures that include NMR spectra.

4/ Why is the region showing the UDP signals not shown? Regarding the presence of Mn²⁺, it would be interesting to search for specific line-broadening effects on UDP-signals. Can this be included?

Response#: Thank you for the comment. The strong PRE effect in UDP signals was previously discussed. Since the signals of UDP are absent in our STD experimental conditions and the main aim was to investigate the binding of the glycopeptide, the region of ¹H-NMR spectrum at low field (region where the chemical shift of uracil protons and H1 of ribose appear) is not displayed in the spectra.

Action#: The following text was included in the experimental section.

“In presence of 150 μM MnCl₂ strong paramagnetic relaxation enhancements are observed for UDP, which prevents the observation of UDP proton signals in the NMR spectra. However, the presence of MnCl₂ does not preclude the observation of the proton signals of the glycopeptides and allows to extract information from STD experiments.”

5/ Minor: The phrasing "different irradiations" should be substituted with "different on-resonance frequencies".

Action#: This has been fixed in the revised version of the manuscript.

Fig. S5: Impurities needs to be marked in all spectra. The H6 protons show rather intense signals. Are the integrals OK? The resolution (pixel quality) of the spectra is rather low.

Response#: We agree with the reviewer and the impurities were identified in all the NMR spectra. We have also verified the signals for the H6 protons and the STD intensities are right. H6 geminal protons often display smaller STD responses due to their higher relaxation rate in comparison to the other protons (see J. Magn. Reson. 2002, 155, 106–118).

Action#: The signal for Tris-buffer and HDO are now displayed in the spectra. Spectra with better resolution were added.

Fig. S6: It would be nice to have the parameters ΔG , ΔH , etc. as an insert.

Action#: This has been included in the corresponding Supporting Figure.

Tables S2-S4: Definition of STD_{AF} .

Response#: We agree with reviewer.

Action#: The table caption of tables S2-S4 were modified to include STD_{AF} . In the experimental section the definition of STD_{AF} was added.

Draft Only

Reviewer #3 (Remarks to the Author):

The rationale of the work is that studies of rat, human and fly C1GalT1 orthologues suggest that the enzymatic activity is affected by the protein sequence. In the absence of a structure of C1GalT1 with bound substrates, the molecular basis for peptide sequence recognition is not known. The key result of the work is the determination of the C1GalT1 structure with bound substrate (peptide + sugar), which shows a strained conformation for the peptide, and allows a description of the catalytic mechanism to be devised.

The authors link their study to the tumour-associated T and Tn antigens. The importance of their work to cancers is tenuous at best without a clear rationale provided in the introduction. Actually, they don't consider the implications of their work to tumours in any detail in their manuscript at all. I suggest removing these connections, especially in title, as it does not reflect the work.

Response#: We have mentioned the strong link between *Cosmc*/C1GalT1 and cancer in the introduction because this is one of the most important aspects of C1GalT1 and its private chaperone. However, and as indicated by the reviewer, we do not bring any implications from our work in cancer because we cannot infer from structural studies how potential mutations in C1GalT1 might affect glycosylation and lead to cancer because most of the cases associated to cancer in which the Tn is present are due to hypermethylation of the *Cosmc* promoter. Due to that, we have preferred to be cautious and avoided any comments related to our findings and their potential implications in the understanding of cancer. However, we still think that it is important to mention briefly the importance of *Cosmc* and C1GalT1 in cancer in the introduction.

Action#: We have removed the word "cancer" from the title as suggested by the reviewer.

Enzyme kinetic studies. The data does not support the claim that "...peptide sequence is crucial for achieving optimal kinetic parameters..." Firstly, the difference in the parameters is not particularly large and probably within error.

Response#: We respectfully disagree with this comment. If you compare the kinetics of α -O-methyl-GalNAc with those of the glycopeptides, it is clear that we cannot reach saturation with α -O-methyl-GalNAc while for the glycopeptides, and in all cases, we can reach saturation, allowing us to get the kinetic parameters. This is already enough to demonstrate that the peptide sequence is required to achieve optimal kinetic parameters because in the absence of the peptide, GalNAc is not enough to determine the kinetic parameters under our conditions. Just also for clarification purposes, our comparison was referred to the presence of the peptide in the context of the glycopeptide compared to the GalNAc moiety itself. Regarding the small differences between our glycopeptides, we know that there are very small differences between glycopeptides, which was also supported by the references cited in the manuscript, where they also found similar small differences among glycopeptides.

Action#: For clarification purposes, we have added the following sentences in the manuscript,

"Overall, our data suggest that the differences in the kinetic parameters between the glycopeptides are small and that not only the GalNAc moiety is important for glycosylation, but also that the peptide sequence is crucial for achieving optimal kinetic parameters, suggesting that C1GalT1 may interact directly with the peptide of the acceptor substrates. Note that saturation is not achieved in the presence of α -O-methyl-

GalNAc and that only this is achieved in the presence of the different peptides within the glycopeptide substrates”.

Secondly, the kinetic parameters determined are ‘apparent’ kinetic constants (and should be labelled as such), since the catalysed reaction is a two substrate reaction and they have kept the concentration of UDP-constant at a sub-saturating concentration. The authors repeat the kinetic experiments varying also the UDP-Gal concentration and fit to ping-pong or ternary complex kinetic models. It is likely the different peptides could subtly alter the K_m for UDP-Gal, and this could explain the perceived differences in the apparent kinetic parameters.

Response#: Thank you for the constructive comment. We have analyzed our conditions for the determination of the K_m for UDP-Gal and have realized that we overestimated the K_m because simply the experiments were not done under saturated conditions. In fact, initially we determined the K_m for UDP-Gal under a sub-saturated concentration of the APDT*RP (fixed at 250 μ M). Due to this, now we have performed further kinetic experiments to determine the K_m of UDP-Gal using saturated concentrations of the APDT*RP and P4. The results showed very similar results and rendered a better K_m for UDP-Gal than the previously determined. Thus, we can certainly conclude that our experiments to determine the K_m s for the peptides were done under saturated concentration of UDP-Gal.

Although we know that there are small differences between our peptides, we also have to say that there is a nice correlation in the differences found between the ITC and the kinetic parameters. It is clear that ITC and kinetic parameters provide different constants, however, it is important to mention that the K_d s for the glycopeptides matched their K_m s and the differences found between the K_m s (~3.5-fold better K_d of P4 than that of APDT*RP). This is important to emphasize it because it supports that the differences between our glycopeptides are trustable and reliable, specially since these experiments were done under a saturated concentration of UDP-Gal.

Finally, we have not performed experiments to determine the type of kinetic models for DmC1GalT1 because these experiments are out of the scope of our manuscript.

Action#: We have determined the K_m of UDP-Gal in the presence of the APDT*RP and P4, rendering a slightly better K_m of UDP-Gal in the presence of P4 than in the presence of APDT*RP. In addition, and based on our K_m s, the experiments to determine the K_m s for the peptides have been conducted under saturated conditions. Due to this, we have not included the term “apparent” for our kinetic parameters. The new data have been incorporated to the main text and also to the SI.

The finding that APDT*RP only binds in the presence of UDP (Fig2b) is very interesting and suggests that the kinetic mechanism is altered (ordered binding vs. random binding)—

Response#: We agree that these differences between glycopeptides is very interesting. However, performing further kinetic experiments to determine the kinetic mechanism is out of scope of our article. In the revised version of our manuscript, we have also provided an explanation regarding why we performed the ITC, STD and computational experiments with and without UDP. This is sufficiently explained and detailed in our manuscript (see below our comments regarding the Hill model).

Action#: No action taken.

Please add error bars to the k_{cat}/K_m plot in Figure 1c.

Action#: The error bars have been added to the k_{cat}/K_m plot in Figure 1c.

NMR experiments. This reviewer is not an expert in NMR. The key finding here is that it is the sugar that drives binding, with only minor contributions by the peptide. Allostery. The enzyme DmC1GalT1T43-Q388 does appear to have a sigmoidal curve (supplementary figure7 purple data). The authors should fit this data to a hill model and compare this to a standard Michaelis-Menten model to determine whether for APDT*RP binding is allosteric. Also, please elaborate on why your results "...imply that DmC1GalT1 does not likely follow an induced-fit mechanism..."

Response#: Although the data were adjusted to a hill model through GraphPad Prism, the results showed that the model was ambiguous and the fitting was poor. For a proper fitting and reliability of the Hill model, low concentration values (nM) are needed to fit the bottom plateau and to calculate the Hill coefficient. The kinetic kit (UDP-Glo™ Glycosyltransferase Assay from Promega) would not be able to detect such low substrate concentrations. Even though, the correlation coefficient (R^2) for the hill model was 0.8226 while for the Michaelis-Menten fitting was 0.9571. Therefore, it is clear that our data fit better to the Michaelis-Menten equation.

Regarding the induced-fit mechanism, we have added a small introduction of our findings with other glycosyltransferases that follow an induced-fit mechanism. In addition, we have also mentioned a further clarification of what this mechanism involves in the sections focused on the STD and ITC experiments.

Action#: We have added the following text in the revised version of the manuscript,

"We conducted these experiments in the presence of absence of the nucleotide because previously we found that other distant GT-A fold GTs such as GalNAc-T2, and the NleB/SseK effector proteins were dependent on the presence of UDP for binding to their protein/peptide acceptor substrates, implying the existence of an induced-fit mechanism^{20,34,35}. Interestingly, in the case of GalNAc-T2, the active conformation of GalNAc-T2, characterized by the shifting of a flexible loop from an open to a closed conformation, was completely achieved in the presence of UDP-GalNAc and less in the presence of UDP²⁰. We also found that the GT-B fold FUT8 showed similar properties to the other two GTs, though FUT8 bound better to an N-glycan in the presence of GDP, and the nucleotide was not essential for binding to the N-glycan³⁶. In addition, NleB/SseK and FUT8 also contained flexible loops that were ordered in the presence of the nucleotide as we found for GalNAc-T2, implying that the active site adopted an active conformation once that these flexible loops bound to the sugar nucleotide^{20,34,35,37}. Overall, we proposed for these enzymes that the binding of the sugar nucleotide was required for binding to the acceptor substrate (optimal binding for FUT8 in the presence of GDP) and in turn for glycosylation".

"...., and that therefore, *DmC1GalT1* does not need prior binding to the sugar nucleotide to bind its acceptor substrates".

The structural analysis is solid, although there are already structures of homologues in the pdb, which perhaps renders the work incremental. The key finding being that the substrate is held in the active site in a strained conformation, but that the substrates are

well placed for an inversion of configuration SN2 mechanism. The residues important for substrate binding and catalysis are further probed using mutagenesis (in vitro and in cell, which is a very nice assay). The MD simulations help explain the different binding patterns seen for the peptides, but otherwise don't add significantly to the work without accompanying experimental data to backup these models of the apo and UDP-Gal structures.

Response#: We respectfully disagree with the comment regarding that our structure is only incremental. This is clearly not the case because as we mentioned in the manuscript, the structures of the homologues published before were those of the human B3GNT2 and Maniac Fringe, two distant glycosyltransferases from C1GalT1. While the human B3GNT2 only recognizes sugars and not peptides, the Maniac Fringe recognizes glycosylated EGFs and in turn likely peptide sequences. In addition, B3GNT2 and Maniac Fringe have completely different acceptor substrates to those of C1GalT1. However, there were not structures of Maniac Fringe complexed to glycosylated EGFs impeding the understanding of how this enzyme recognizes peptide sequences or achieves glycosylation. Therefore, our structural findings are not incremental since they have help us to figure out the catalytic and recognition mechanism of C1GalT1 on glycopeptides, and understand how this enzyme recognizes α -GalNAc-Thr and α -GalNAc-Ser. This is the first time that a structure of this family of enzymes, and also from the family GT31 in the CAZy database, has been solved complexed to a glycopeptide.

Action#: No action taken.

Other comments.

1) for the statement “Nevertheless, it is predicted that C1GalT1 efficiently transfers Gal to all GalNAc moieties (Tn) on proteins indiscriminately and independently of the underlying peptide sequences and clustering of GalNAc O-glycans” please provide a reference. Or, explain why you predict this.

Response#: The reviewer is right and in the revised version of the manuscript we have added a reference that illustrates that the GalNAc O-glycoproteome is vast and with enormous sequence variation around glycosites, implying that C1GalT1 must efficiently transfer Gal to all GalNAc moieties on proteins.

Action#: We have added a new reference (PMID: 23584533) and changed the previous sentence as follow,

“Nevertheless, the GalNAc O-glycoproteome is vast and with enormous sequence variation around glycosites, so it is predicted that C1GalT1 efficiently transfers Gal to all GalNAc moieties (Tn) on proteins indiscriminately and independently of the underlying peptide sequences and clustering of GalNAc O-glycans²⁷”.

2) In figure 3b, it is not clear what this density is. As stated it is difference density? Usually difference density is red (-3 sigma) and green (+3 sigma). Perhaps it is from an omit experiment?

Response#: The electron density map corresponds to the Fo–Fc (blue) map contoured at 2.2 σ for APDT*RP and UDP as indicated in the legend of Figure 3b.

Action#: No action is required because this was already indicated in the legend of Figure 3b.

3) The link to Cosmc driving oligomerisation is not substantiated by the data and should be removed. (lines 221-227)

Response#: We respectfully disagree with the comment and we think that it is interesting to speculate that Cosmc might help in the oligomerization of C1GalT1. Cosmc binds to a Leu that is located in the interface so therefore it is tempting to speculate that one of the functions of Cosmc could be to drive the dimerization of C1GalT1.

Action#: No action taken.

4) Figure 4, a ligPlot would aid the reader.

Response#: We agree with the reviewer. We have tried LigPlot and other programs to render a scheme containing the interactions. However, we think that it is better a stereo figure of the original figure in which we have decreased the size of the sticks and the sphere for the Mn^{2+} ion.

Action#: Fig 4a depicts a stereo figure of the active site that facilitates the visualization of the interactions between *Dm*C1GalT1 with UDP, Mn^{2+} and the glycopeptide.

5) The introduction should provide more foundational/basic information to help orient the reader and to broaden the readership—for example a figure showing T and Tn antigen synthesis would help.

Response#: We agree with the reviewer.

Action#: We have added a figure of the synthesis of T and Tn and other types of core O-glycans in Fig. 5a. We think that the scheme in this figure is the right place to understand the text in which the enzymes and the different pathways are mentioned. In addition, we have enlarged the introduction to have more basic information on C1GalT1 and its role in biology, see below for the added text,

“The C1GalT1 is unique among metazoan GTs in that its folding, stability and activity only in higher eukaryotes depends on a private X-linked chaperone Cosmc⁸, which interestingly exhibits sequence similarity with C1GalT1 and lacks the catalytic DxD motif⁹. Interestingly, in lower eukaryotes such as *Drosophila melanogaster* or *Caenorhabditis elegans*, C1GalT1 related sequences also exist^{5,10}, but these enzymes do not appear to require a chaperone for expression¹¹. The endoplasmic reticulum Cosmc binds to the unfolded C1GalT1 and is required for folding of C1GalT1⁶. Both C1GalT1 and Cosmc are ubiquitously expressed, which corresponds with the detection of core 1 O-glycans structures in most cells^{8,12,13}. *C1GalT1* homozygous knockouts (KOs) in mice and *D. melanogaster* exhibit embryonic lethality, with defective angiogenesis and fetal embryonic hemorrhage in mice, and a predominant central nervous system phenotype in *D. melanogaster*, indicating that O-glycosylation is essential for normal development and angiogenesis^{10,14}. The functions of C1GalT1 and Cosmc have demonstrated that O-glycans may conceivably interact with almost all physiological processes, including tissue homeostasis, our immune system homeostasis, the homing and circulation of our blood cells, the protection and integrity of inner and outer epithelial barriers, and maintenance of B cell tolerance¹⁵⁻¹⁷. Regarding tumorigenesis and metastasis, the Tn antigen is highly expressed in human solid tumors, being one of the most recognized TACAs. In most cases, the Tn antigen is formed due to the hypermethylation of the Cosmc promoter leading to its silencing¹⁸. Aberrant Tn expression is associated with oncogenic features, including proliferation, migration, and invasion of cancer cells^{6,7}. The

silencing of Cosmc has also been used to glycoengineer HEK or CHO cells to produce SimpleCell lines allowing the interrogation of the activity of GalNAc-T isoenzymes and analysis of the functions of protein glycosylation¹⁹”.

6) Figure 4e, the mechanism is missing an arrow, a charge is incorrect, and the transition state has a covalent bond that should be a transitional/breaking bond.

Action#: This has been fixed in the revised version of our manuscript.

Editorials:

Line 64 extend should be extent?

Action#: This has been fixed in the revised version of our manuscript.

Line 299 present should be predicted

Action#: This has been fixed in the revised version of our manuscript.

Line 337 Arg125 should be Arg152?

Action#: This has been fixed in the revised version of our manuscript.

Figure 3c – what does the orange colour denote?

Response#: The orange color denotes high conservation between C1GalT1 orthologues while the white color denotes dissimilar or variable residues between orthologues.

Action#: No action taken.

Draft Only

Reviewer #4 (Remarks to the Author):

The manuscript characterizes the interaction between various glycopeptides with the galactose transferase C1GalT1. Apart from kinetic, NMR, ITC and crystallographic studies, the experiments are complemented by molecular dynamics simulations of various complexes. I will focus my review on the molecular simulations. One of the key findings from the simulations is the observation that for the free glycopeptide, the Psi angle of the glycosidic bond is preferably found at an angle of 120 degree, while in the crystal structure it is seen at 180 degree. This is an interesting finding, corresponding to a conformational selection model and possibly explaining why the enzyme is also active on Ser-glycosylated peptides, in which the 180 degree angle is preferred also for the free peptide. The work seems carefully done and reads well.

I would like to make the following comments:

1. The observed conformational preferences for the Thr-peptide, could be interpreted by a conformational selection model on the substrate. In line 311, the authors refer to the conformational penalty as an entropic penalty, which is apparently overcome by favorable interactions. I do not quite agree that this effect is purely entropic; energetic contributions will also contribute to the preference of the free peptide. In fact, the authors should be able to quantify the free-energy penalty of bringing Psi from 120 to 180 degree by direct counting the conformation with angles > 150 and < 150 degree. From [doi: 10.1002/cphc.201900079], I estimate this value to be about 10 kJ/mol. A proper integration over the conformational regions of interest as in [doi: 10.1021/acs.jcim.7b00351] could possibly reduce this value. The authors should attempt to quantify the preferences from the simulations of the free peptide.

Response#: We thank the reviewer for highlighting this important aspect.

Action#: Following the suggestion, we have calculated the free energy free-energy maps of glycopeptides **APDT*RP** and **APDS*RP** in water at 300K and 1 atm, as a function of the glycosidic dihedral angles (ϕ, ψ). To this purpose, we have used the Adaptively Biased Molecular Dynamics (ABMD) method, which is implemented in AMBER 20 (V. Babin, C. Roland, and C. Sagui, "Adaptively biased molecular dynamics for free energy calculations", J. Chem. Phys. 128, 134101 (2008); V. Babin, V. Karpusenka, M. Moradi, C. Roland, and C. Sagui, "Adaptively biased molecular dynamics: An umbrella sampling method with a time-dependent potential", Int. J. Quant. Chem. 109, 3666 (2009); M. Moradi, V. Babin, C. Roland and C. Sagui, "The adaptively biased molecular dynamics method revisited: new capabilities and an application", J. Phys. Conf. Ser. 640, 012020 (2015)). A brief description of these computational calculations has been included in the section 'MD simulations protocol'.

The free-energy map of the glycopeptide **APDT*RP** has been included in Figure 4c. In addition, a **Supplementary Figure 10** (see below), showing the maps for both glycopeptides (to facilitate the comparison), has been included in the revised version of the Supplementary Information file.

Supplementary Figure 10. Free-energy maps (ϕ , ψ) of the glycosidic dihedral angles calculated for the free peptides in water and using the Adaptively Biased Molecular Dynamics (ABMD) method implemented in AMBER 20 (see Methods) at 300 K. The contour maps are drawn with a spacing of 1 kcal/mol. Regions that were never visited by the peptides are shown in dark orange. “A” refers to the ‘eclipsed’ conformation typically found for α -GalNAc-Thr derivatives in solution^{1,2}. “B” refers to the ‘staggered’ conformation found for α -GalNAc-Ser derivatives in solution³.

According to these calculations and in good agreement with the reviewer's estimates, the free energy penalty to bring Psi from 120 (conformation A in figure shown above) to 180 (conformation B in figure shown above) degrees in the APDT*RP compound is 2.5 kcal/mol (≈ 10.5 kJ/mol). In contrast, this conformational shift is favored by 1.9 kcal/mol (≈ 7.9 kJ/mol) in the serine derivative.

2. As the APDS*RP peptide shows slightly worse affinity, and this peptide will not have the conformational penalty mentioned in the previous point, the conformational penalty of APDT*RP should be compensated by the transfer of the hydrophobic CH₃ group in Thr from water to a favorable CH- π interaction. The authors could emphasize that rather subtle free-energetic effects probably guide the binding.

Response#: We thank the reviewer for highlighting this important aspect.

Action#: This aspect has been now discussed in the revised version of the manuscript. The new paragraph reads as follows:

“...the hydrogen bonding between the side chain of Tyr304 and GalNAc OH6 was lost. On the other hand, APDS*RP peptide has slightly worse K_m than the threonine derivative and does not have the conformational penalty that operates in the Thr- containing peptide. These results suggest that rather subtle free-energetic effects are probably guiding the binding. In this regard, the free energy penalty associated to bring the glycosidic linkage from a ‘eclipsed’ conformation to a ‘staggered’ one was calculated to be 2.5 kcal/mol (Fig. 4c and Supplementary Fig. 10). In contrast, this conformational shift is favored by 1.9 kcal/mol in the serine derivative (Supplementary Fig. 10).

3. I would encourage the authors to provide a table with a clear overview of the

interactions determined by the STD NMR experiments and the observations in the MD simulations. Are all the relevant atoms indeed in close proximity?

Response#: We thank the reviewer for the important suggestion.

Action#: Two supplementary tables (**Tables 7 and 8**) were included in new version of Supporting information. As well, we added the text below on the main manuscript in the section of MD simulations.

In the context of APDT*RP, the new text reads as follows:

“In addition, the binding mode for the glycopeptide observed by MD simulations agrees with the STD experiments described above (**Supplementary Table 7**).

In the context of P4, the following text has been added in the revised version of the manuscript:

“Also, in the case of glycopeptide P4, good agreement is observed between the glycopeptide-protein interproton distances derived along the MD simulations in the presence of UDP-Gal and the STD responses estimated for GalNAc, Pro1, Ala2, and Thr4 (**Supplementary Table 8**). Transient close contacts between Ala3 or Tyr7 with protein residues were observed throughout the MD trajectory, which could also explain the STD response for these amino acids”.

4. Can the authors say more about the failed crystallography experiments with the other glycopeptides (page 9)? This seems to be in contrast to the tighter binding and higher activities for these peptides?

Response#: We cannot really say much about that comment. We tried to get the structure of *DmC1GalT1* with other glycopeptides but we could never get crystals despite several attempts. And we know that some of these glycopeptides bound slightly better. We guess that this is part of the uncertainty behind protein X-ray crystallography.

Action#: No action taken.

5. Some minor textual glitches:
a. Line 64: extend -> extent

Action#: We have fixed this mistake in the revised version of the manuscript.

b. Line 140: We performed then -> We then performed

Action#: We have fixed this in the revised version of the manuscript.

c. Line 191: these somewhat were complex -> these were somewhat complex

Action#: We have fixed this in the revised version of the manuscript.

d. Lines 291, 298: *DmC1GaT1* -> *DmC1GalT1*

Action#: We have fixed this in the revised version of the manuscript.

e. Lines 326 to 328: The activity ... was completely inactive -> Rephrase

Action#: We have fixed this in the revised version of the manuscript. The readability of that sentence has been improved by removing part of the original sentence.

REVIEWERS' COMMENTS

Reviewer #2 (Remarks to the Author):

The authors have fixed all issues raised. To my opinion, the manuscript is now essentially ready for publication. Nice work!

Reviewer #3 (Remarks to the Author):

The authors have largely responded adequately to my comments.

The only remaining issue is that they must call the kinetic parameters (K_m and k_{cat}) apparent parameters (K_{mapp} and k_{catapp}), since full kinetic analyses (i.e. varying both substrates) was not done.

Reviewer #4 (Remarks to the Author):

The authors have addressed all of my comments satisfactorily. I have no further reservations towards publication of this manuscript.

REVIEWERS' COMMENTS

Reviewer #2 (Remarks to the Author):

The authors have fixed all issues raised. To my opinion, the manuscript is now essentially ready for publication. Nice work!

Response:
Thank you.

Reviewer #3 (Remarks to the Author):

The authors have largely responded adequately to my comments.

The only remaining issue is that they must call the kinetic parameters (K_m and k_{cat}) apparent parameters (K_{mapp} and k_{catapp}), since full kinetic analyses (i.e. varying both substrates) was not done.

Response:
Thank you.
In in the revised version of our manuscript, apparent k_{cats} and K_{ms} are now mentioned in the manuscript and the Supplementary Information. In addition, figure 1 has been updated to show the apparent k_{cats} and K_{ms} . The changes are highlighted in blue.

Reviewer #4 (Remarks to the Author):

The authors have addressed all of my comments satisfactorily. I have no further reservations towards publication of this manuscript.

Response:
Thank you.